# Fusing subnational with national climate action is central to decarbonization: the case of the United States

Nathan E. Hultman [1✉], Leon Clarke[1], Carla Frisch[2], Kevin Kennedy[3], Haewon McJeon[1], Tom Cyrs[3], Pete Hansel[4], Paul Bodnar[2], Michelle Manion[3], Morgan R. Edwards [1,7], Ryna Cui[1], Christina Bowman[1], Jessie Lund[2], Michael I. Westphal [3,5], Andrew Clapper [6], Joel Jaeger[3], Arijit Sen[1], Jiehong Lou[1], Devashree Saha[3], Wendy Jaglom[2], Koben Calhoun[2], Kristin Igusky[3,8], James deWeese [3,9], Kareem Hammoud[2], J. C. Altimirano[3], Margaret Dennis[3], Chris Henderson[3], Gill Zwicker [3] & John O'Neill[1]

Approaches that root national climate strategies in local actions will be essential for all countries as they develop new nationally determined contributions under the Paris Agreement. The potential impact of climate action from non-national actors in delivering higher global ambition is significant. Sub-national action in the United States provides a test for how such actions can accelerate emissions reductions. We aggregated U.S. state, city, and business commitments within an integrated assessment model to assess how a national climate strategy can be built upon non-state actions. We find that existing commitments alone could reduce emissions 25% below 2005 levels by 2030, and that enhancing actions by these actors could reduce emissions up to 37%. We show how these actions can provide a stepped-up basis for additional federal action to reduce emissions by 49%—consistent with 1.5 °C. Our analysis demonstrates sub-national actions can lead to substantial reductions and support increased national action.

[1] Center for Global Sustainability, 2101 Van Munching Hall, School of Public Policy, University of Maryland, College Park, MD 20742, USA. [2] Rocky Mountain Institute, 2490 Junction Place, Suite 200, Boulder, CO 80301, USA. [3] World Resources Institute, 10 G Street NE, Ste. 800, Washington, DC 20002, USA. [4] Independent, Washington, DC, USA. [5] Basque Centre for Climate Change (BC3), Edificio Sede 1-1, Parque Científico de UPV/EHU, Barrio Sarriena s/n, 48940 Leioa, Spain. [6] CDP North America, 127 West 26th Street, Suite 300, New York, NY 10001, USA. [7]Present address: La Follette School of Public Affairs, 1225 Observatory Drive, University of Wisconsin–Madison, Madison, WI 53706, USA. [8]Present address: U.S. Climate Alliance, 1750 Pennsylvania Avenue NW, Suite 300, Washington, DC 20006, USA. [9]Present address: Transportation Research at McGill (TRAM) Centre, School of Urban Planning, McGill University, Macdonald-Harrington Building, 815 Rue Sherbrooke Ouest #400, Montréal, QC H3A 0C2, Canada. ✉email: hultman@umd.edu

A global transition to avoid the most severe climate risks will require a "rapid and unprecedented societal transformation[1]." Yet national governments have not undertaken actions at a sufficient pace and scale, nor with sufficiently robust political support, to deliver the necessary actions[2–6]. Against this backdrop, an increasing number of actors—states, cities, provinces, businesses, and more—are taking significant measures to reduce emissions, and such actions are now becoming sufficiently broad and ambitious to challenge conventional understanding of both the models for achieving global success and the scale of their potential impact. While climate policy has often been perceived as driven by national governments—particularly by governments themselves—the potential value of this "bottom-up" action in delivering higher global ambition to reach 1.5 °C compatible pathways may be much greater than has been realized.

This principle of diversified, non-state action has been noted by advocates, the research community, and sub-national leaders as an important basis for climate policy for decades and formed the basis for organizing coalitions of climate actors even well before the Paris Agreement[7,8]. It was then embedded in the preamble of the Paris Agreement itself, which "recogniz[es] the importance of engagements of all levels of government and various actors[9]." After Paris, such activities expanded in number and scope, underscoring the potential impact from a broader engagement of these actors[7,10–12]. For example, at the international level, many cross-border, non-state, and multi-stakeholder initiatives have emerged—focusing on carbon neutrality, science-based targets, elimination of coal-fired electricity, waste, industrial efficiency, and many more[13,14]. Such initiatives could reduce emissions up to 15–18 Gt $CO_2$e/yr beyond national commitments alone[15]. At the same time, in national domestic contexts, a rapidly growing number of non-national governments and other actors have been committing to increasingly ambitious and robust actions on climate. A recent assessment of the impact of non-state action drew on a broad international data collection strategy and a global integrated assessment model to show that such actions could deliver reduced emissions at the global level[16].

Nevertheless, our understanding of both the impacts on national emissions trajectories and the policy platforms that would enable those reductions has been limited. Answering such questions and building a more robust global capacity to assess and integrate the full scope for contributions from bottom-up actors is a key step toward unlocking the full potential for such action. Because of the imperative to better understand the potential for rapid scale-up of emissions reductions by these actors[13,17], methodologies are evolving rapidly. Such estimates must account for different kinds of targets and policies at diverse scales. However, because of the relatively fine scale of sub-national commitments, consistent and high-quality data availability has been a challenge, and the methods of integrating them into national-level projections are still underdeveloped.

In this paper, we focus on a key global emitter—the United States—and describe how a significant subset of those actors, including states, cities, and businesses, are implementing policies that will significantly reduce economy-wide emissions. Using new methods to aggregate and analyze the many overlapping actions by cities and states in the United States, we quantify the potential scope and implications of current actions. We also evaluate the potential impacts of a more widespread application of the examples set by the most ambitious actors. The analysis also demonstrates that expanded action from those actors can provide an increased starting level of policy ambition, and a set of potential policy options, for additional federal action. These new methods for aggregation and analysis can serve as an example for assessing potential impacts of bottom-up action and thus developing roadmaps to higher ambition in diverse national contexts.

## Results

**A case of multi-level climate action: national, state, city, and business actions in the United States.** The United States today provides a globally critical testing ground for essential questions on how subnational commitments can contribute to better outcomes for overall national climate strategies. Recent reversals in U.S. climate policy at the national level have dominated headlines —including the initiation of withdrawal from the Paris Agreement and numerous deregulatory actions driven by the Executive branch. Nevertheless, the U.S. Federal political system devolves numerous authorities to sub-national and non-governmental actors, creating opportunities for multi-level policy implementation. This multi-level governance approach has long been central to U.S. energy system dynamics and other areas of governance. It has more recently been identified as important in the context of climate action, emerging in tandem with the rise of climate change as a policy priority and in particular during previous periods when national-level climate leadership has been weaker.

The long-term evolution of state policies, in particular, has consistently pointed toward their potential for driving national emissions reductions even while national climate policies were absent, scattered, or delivered without national legislative approaches[18–20]. For example, the national government can establish laws, regulations, and standards governing energy efficiency, renewable fuels, air quality, vehicle fuel economy, research and development, and many other areas. But states also have significant authority over energy and transportation systems —for example, they can mandate renewables in the electricity sector through net-zero energy policies, renewable portfolio standards, or clean energy standards. In some cases, they also set standards for vehicle emissions and fuel standards, and some transportation and land policies. Likewise, cities and counties have the authority to set and monitor the implementation of zoning, land use, and building codes. In addition to such policy authorities, private sector firms, investors, and other non-governmental organizations are largely unconstrained in their ability to make investment and procurement decisions that reflect climate considerations[21]. While the aggregate effects of subnational action may serve to build a higher overall level of action in the United States, individual sub-national actions are not always consistent with long-term climate goals or clean energy transitions; they may also lead to conflicts and economic inefficiencies[22]. States have sometimes also opposed federal climate policies through court actions. Also important is the fact that the U.S. political system is federal; we might expect more potential impact from sub-national actions in federal systems compared to systems that do not devolve authorities in the same way.

Despite this tension, non-federal action in the U.S. has consistently served as a primary, and sometimes the sole driver of climate action. Over the last several years, federal policy reversals have spurred support for mitigation policies in many places[23], leading to new commitments at a scale not before seen. This expansion has occurred in several dimensions: the number of entities committing to climate action (Fig. 1a), the breadth of their activities, the ambition of their commitments, and the robustness with which these commitments are embedded into policy processes[24]. Coalitions of non-federal policy actors have formed to reflect and enhance their own communities' or constituencies' interest in climate action. For example, the "We Are Still In" coalition[25] includes nearly 4000 U.S. states, cities, businesses, universities, communities of faith, tribal groups, and

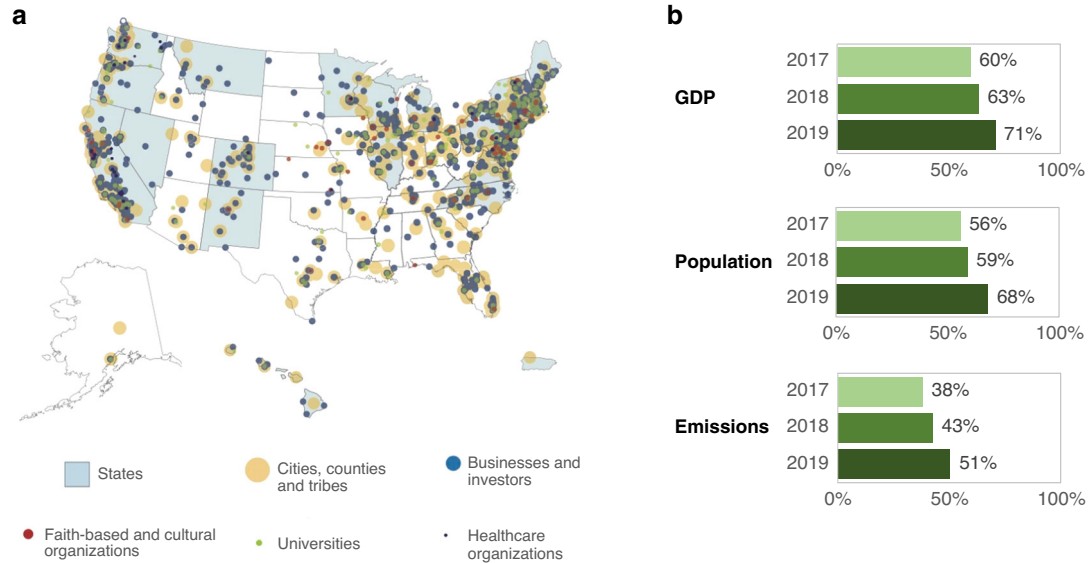

*Figure reproduced with permission from Hultman et al., Accelerating America's Pledge (25)*

**Fig. 1 The group of U.S. coalitions of non-federal actors making commitments to climate goals is large, growing, and globally significant. a** States, cities, businesses, universities, faith-based, and other organizations now number greater than 3800[24]. This study assessed the most significant subset of the actors shown in the map, focusing on states and the 285 largest cities. **b** Growth of the coalition from 2017–19, now representing the equivalent of the world's second-largest economy (after the entire United States).

more. Other coalitions have also emerged, such as the bipartisan U.S. Climate Alliance which now includes 25 governors committed to meeting the goals of the Paris Agreement[26]. Moreover, this active and expanding bottom-up political engagement on climate and energy policy[27] can potentially impact emissions at the national level.

The scale of sub-national coalitions can be estimated in several ways, such as their share of national economic activity, population, and current greenhouse gas (GHG) emissions. We estimated these scale metrics in the United States and found that these U.S. coalitions of actors are globally significant (Fig. 1b). States and cities with these commitments collectively represent 65% of the U.S. population, 68% of U.S. GDP, and 51% of U.S. emissions[24]. Were they a country, they would constitute the world's second-largest economy (after the entire United States), approximately the size of China's, and the fourth-largest GHG emitter.

**Assessing the current and potential impact of sub-national climate action.** In light of this groundswell of diverse new climate commitments, the research presented here estimates current and potential impacts from commitments and actions taken across all 50 U.S. states, the 285 largest cities (those with populations over 100,000 or that have readily-available data on climate commitments), and a set of corporate actors (Supplementary Methods). Estimating aggregate emissions impacts from these actions is complex. This process includes familiar challenges for all economy-wide emissions projections, including but not limited to appropriate scenario construction and ensuring robust modeling that includes all major GHGs, and appropriate ecosystem and economic interactions. Additional challenges arise in understanding and projecting forward the potential impacts from policy units at sub-national scales[28]. For example, baseline data are frequently heterogeneous or incomplete, an analysis must consistently and transparently incorporate different kinds of targets operating at different scales and with potentially different levels of implementation, and commitments must be aggregated in ways that avoid double-counting of potential emissions reductions. Finally, assessing the potential for non-federal actors

to catalyze new action also requires a detailed scoping of opportunities for enhanced policies.

To address these challenges, we developed a new, comprehensive strategy to assess the scope and emissions impact of these non-federal climate actions in the United States. We do this by integrating data on baselines and sector-specific policies and commitments into an economy-wide estimate of the U.S. emissions trajectory to 2030. The overall process involved the steps of data collection, baseline development, aggregation, and modeling. We also conducted parallel stakeholder engagement to help develop the policy platform for these scenarios. The resulting approach combines a detailed accounting of the most significant bottom-up actions with an integrated assessment model to understand their overall emission impacts. The approach is based on an existing open-source, global IAM with a 50-state resolution, GCAM-USA, which we modify as GCAM-AP for this study[29]. Because the resolution of IAMs is too coarse to incorporate the detailed impacts of diverse non-federal action, we developed a methodology to aggregate data on current sub-national measures to interact with GCAM.

In particular, the aggregation process draws from a well-established set of inputs from GCAM and then aggregates the effects of diverse policies affecting emissions while explicitly considering the overlap between actors and avoiding double-counting within each sector. Our aggregation process is first calibrated to the GCAM-AP baselines. Then, using a structured methodology (Supplementary Methods), we determine what policies are included in our analysis, what their impact will be, and the degree to which they generate activity that goes beyond what the previous policy architecture would deliver. This allows us to consistently estimate impacts at the sector level, in terms of activity data appropriate to that sector (for example, TWh of renewable generation from state and municipal policies, number of zero-emission vehicles required to meet state and local procurement, or HFC emissions reductions from state and corporate initiatives). The sector-level analysis and projections make use of historical emissions data, activity data, and policy or target information from a range of data sources. GCAM then uses

**Table 1 Overview of measures evaluated for the U.S. current measures scenario (left) and impact in 2030 (right).**

| Type of action | Specific measures evaluated | Projected impact if achieved |
|---|---|---|
| Renewable mandates | Binding renewable portfolio standards in 28 states | Renewable generation increases to 26% of total generation by 2030 |
| Renewable goals | Significant non-binding renewable goals in 6 states, commitments in 142 cities, and recent renewable energy and/or decarbonization commitments from 24 utilities | Not included in the current measures scenario (achievement of these actions is modeled in the enhanced non-federal and comprehensive scenarios) |
| Retirement of coal-fired power plants | Coal plants continue to retire according to announced and scheduled retirements and projected closures of additional uneconomic units | Coal falls to 16% of total generation by 2030 from 27% in 2018 |
| Nuclear fleet retention | Policy action in Connecticut, Illinois, New Jersey, New York, and Ohio prevents at-risk plants from retiring | Nuclear generation supplies 17% of generation in 2030 |
| Regulation of fugitive oil and gas operations | Regulations limit fugitive emissions through equipment standards for new and existing facilities in seven states; Federal standards to limit emissions from new facilities are also assumed to remain in effect but at 75% effectiveness | Cumulative 995 Mt $CO_2e$ avoided emissions (2020–2030) |
| Power sector carbon caps | Participation in Regional Greenhouse Gas Initiative (RGGI) by nine Northeast states | Cumulative 160 Mt $CO_2e$ avoided emissions (2020–2030) |
| Voluntary mitigation of fugitive emissions from oil and gas operations | Voluntary mitigation actions on the part of oil and gas companies through EPA's GasStar to limit methane losses | Not included in the current measures scenario (achievement of these actions is modeled in the enhanced non-federal and comprehensive scenarios) |
| Energy efficiency mandates | Binding energy efficiency resource standards (EERS) in 20 states | Cumulative electricity savings of 1566 TWh and gas savings of 2360 BCF (2020–2030), or 541 Mt $CO_2e$ in avoided emissions |
| Energy efficiency goals | Non-binding standards in seven states and efficiency targets in 40 cities | Not included in the current measures scenario (achievement of these actions is modeled in the enhanced non-federal and comprehensive scenarios) |
| Zero-emission vehicle (ZEV) mandates | Current zero-emission vehicles (ZEV) mandates in 10 states requiring minimum share of light-duty vehicle (LDV) sales to be zero emissions | Cumulative total electric vehicle (EV) sales (battery EV + plug-in hybrid EV) of 13.5 million (2020–2030), or 139 Mt $CO_2e$ in avoided emissions |
| Electric vehicle procurement goals | Procurement targets to electrify public fleets in 13 major cities | Not included in the current measures scenario (achievement of these actions is modeled in the enhanced non-federal and comprehensive scenarios) |
| Vehicle emissions standards | States and automakers adopt California's clean cars compromise ensuring incremental vehicle improvements through 2025 | New conventional cars achieve on-road efficiency of 42 miles per gallon by 2025 and remain at that level through 2030; new conventional light-duty trucks achieve 32 miles per gallon by 2025 |
| Regulations to mitigate HFC emissions | Regulations designed to phase down and replace HFCs with low-GWP alternatives in California, Vermont, and Washington and federal standards to limit leakage from refrigerants (EPA Sec. 608) | Cumulative 160 Mt $CO_2e$ avoided emissions (2020–2030) |
| Voluntary mitigation of HFC emissions | Voluntary mitigation actions on the part of supermarkets to reduce HFC emissions through EPA's GreenChill program | Not included in the current measures scenario (achievement of these actions is modeled in the enhanced non-federal and comprehensive scenarios) |
| Maintenance of land sink | No specific actions evaluated for the current measures scenario | Land sink is assumed to remain at current levels (−714 Mt $CO_2e$) through 2030 |

the sectoral level outputs from our aggregation to assess the economy-wide emissions impacts across the United States. The use of this integrated framework reflects global interactions among sectors (Supplementary Methods).

Using this interlinked aggregation method with the GCAM-AP model, we projected emissions impacts under three scenarios. The first scenario, which we call Current Measures, evaluates what existing policies could deliver without additional action. This projection includes impacts from current, already enacted federal and non-federal policies and actions such as renewable portfolio standards, cap-and-trade programs, announced coal plant retirements, state methane reduction policies, and many others. This scenario includes city, state, and business actions as noted above. To keep modeling conservative and analytically rigorous, the scenario did not include commitments to reduce GHG emissions unless they are backed by a binding mechanism. A summary of these policies is provided in Table 1 and further details are in the Supplementary Methods.

We then created two higher-ambition scenarios that assess the impacts of scaled-up actions, one that includes the only action from non-federal actors (which we call Enhanced Non-federal), and another that includes both non-federal and federal action (which we call Comprehensive). These scenarios were constructed to assess the potential for reductions based on sub-national action under futures of high ambition with an increased level of political support and technology progress. A summary of these policies is provided in Table 2, and further details are in the Supplementary Methods.

The enhanced non-federal scenario identifies already implemented and readily expandable actions spanning the major emissions sectors of the U.S. economy and then scales these actions up using three levels of potential ambition. For our enhanced estimates, our approach seeks to estimate the potential for action based on city, state, and business actions using a three-part tiering of states as a proxy. For each sector, we worked closely with key stakeholders and reviewed existing research to

**Table 2 Overview of assumptions for the higher-ambition U.S. scenarios (enhanced non-federal and comprehensive).**

| Policy | Enhanced non-federal scenario 2030 assumptions | Comprehensive scenario 2030 assumptions |
|---|---|---|
| Clean electricity | States establish ambitious clean electricity or renewable portfolio standard requirements. Tier 1, 2, and 3 states reach 60%, 40%, and 20% renewables, respectively. States prevent some at-risk nuclear plants from retiring. 2030 Impact: clean electricity provides 61% of total national generation; renewable energy 40%; nuclear 17% | Federal clean electricity standard and tax incentives. 2030 Impact: clean electricity is 77% and renewable energy is 49% of total generation |
| Fossil fuels | With most coal plants unprofitable, significant coal generation is phased out except in a few holdout states. Market trends and advocacy reduce coal generation nationally. Tier 1 and 2 states constrain new gas plant builds. 2030 Impact: coal produces 7% and gas without carbon capture utilization and storage (CCUS) provides 32% of total generation | Federal policies result in near-complete phase out of coal generation by 2030 and cause gas generation to peak before 2025 and then decline. 2030 Impact: essentially no remaining coal generation; conventional gas produces only 23% of total generation; gas with CCUS produces 12% of generation |
| Oil and gas methane | Tier 1 and 2 states adopt regulations covering new and existing sources, reducing fugitive methane emissions by more than 50%. Tier 3 states achieve reductions where policies are already in place or under development. 2030 Impact: oil and gas methane emissions are reduced by 34% | Federal methane rules are reinstated and strengthened to cover new and existing sources. Fugitive methane emissions are reduced by 60% nationwide. 2030 Impact: oil and gas methane emissions are reduced by 60% |
| Buildings | In Tier 1 and 2 states, all new buildings are 100% electrified; policies are in place to ensure that almost all replacements of appliances from 2030 on are electrified. Tier 1 and 2 states enhance energy efficiency resource standards, with Tier 1 achieving 2% annual savings and Tier 2 achieving 1.5% annual savings. 2030 Impact: Total direct emissions in the buildings sector reduced by 28% compared to 2005 | Due to new federal standards and policies, all new buildings are 100% electrified and replacement appliances from 2030 onward are electrified. Federal financing for residential and commercial retrofits. All states achieve further levels of energy savings where economic. 2030 Impact: total direct emissions in buildings reduced by 31% compared to 2005 |
| Transportation | Tier 1 and 2 states implement zero-emission vehicle mandates and incentives. EVs (BEVs + PHEVs) reach 61% of light-duty vehicle sales. Heavy-duty electric vehicles comprise 15% of new sales in 2030 in Tier 1 and 2 states. In Tier 2 states, adoption is slightly lower. Federal rollbacks of LDV standards prove unsuccessful through 2025. Tier 1 and 2 states set ambitious new vehicle standards post-2026, improving internal combustion engine efficiency by 4% annually. 2030 Impact: total liquid fuel demand from transportation is down 21% from 2005 levels; cumulative EV sales (2020–2030) reach 62 million vehicles nationwide | Federal policies and standards promote zero-emission vehicles so that nationwide EVs (BEVs + PHEVs) reach 62% of new light-duty vehicle sales and 100% of bus sales. The federal government reinstates the current LDV standards through 2025 and improves internal combustion engine efficiency 4% annually from 2026 to 2030. Furthermore, the federal government incentivizes the removal of old and inefficient vehicles from the road. 2030 Impact: total liquid fuel demand from transportation is down 22% from 2005 levels; cumulative EV sales (2020–2030) reach 64 million vehicles nationwide |
| Industry | Tier 1 states, and to a lesser degree Tier 2 and 3 states, incentivize industrial facilities to adopt best-in-class energy management practices and adopt electrified technology. States promote CCUS for industrial uses. Tier 1 and 2 states adopt policies to phase down HFCs and reduce leaks as agreed in the global Kigali Amendment. Tier 1 states adopt standards targeting cement emissions. 2030 Impact: total direct $CO_2$ emissions in the industrial sector are reduced to 5% below 2005 levels; HFCs and other fluorinated gases are reduced to 6% below 2005 | Federal incentives lead all industrial facilities nationwide to adopt best-in-class energy management practices, and federal investments increase the adoption of electrified technology. Federal policies and incentives promote the adoption of CCUS. All states adopt policies to phase down HFCs and reduce leaks. 2030 Impact: total direct $CO_2$ emissions in the industrial sector are reduced to 7.5% below 2005 levels; HFCs and other fluorinated gases are reduced to 37% below 2005 levels |
| Land use | Tier 1 states and some Tier 2 states incentivize low-cost natural climate solutions such as natural forest management, optimal nutrient application, and the use of cover crops. All states mitigate agricultural methane and nitrous oxide emissions where it is cost-effective. 2030 Impact: land carbon sink improves by about 80 Mt $CO_2e$, 11% higher than today | Federal investments and incentives promote low-cost natural climate solutions nationwide. Strong federal incentives promote methane biodigesters to reduce methane from livestock. 2030 Impact: land carbon sink improves by about 167 Mt $CO_2e$, 23% higher than today; livestock $CH_4$ reduced by 29% from the reference case |
| Carbon caps | Tier 1 states meet their legislated economy-wide emissions reduction goals and partially meet their aspirational goals. 2030 Impact: emissions at the national level (all states) reduced by 10% | Tier 1 states meet their legislated economy-wide emissions reduction goals and fully meet their aspirational goals. 2030 Impact: emissions at the national level reduced by 11% |

identify policy tools that would yield significant benefits and that non-federal actors could implement the near-term. In parallel, we generated three groupings of states that reflect the likelihood of accelerated action, based on attributes such as membership in climate organizations, vocal leadership in support of climate action, ambitious emissions reduction targets or standards, and on-the-books policies. In our scenario, Tier 1 states realize levels of ambition roughly equivalent to today's leading states like California and New York, Tier 2 states lag but still move toward the level of ambition of Tier 1, and Tier 3 states do not implement major changes (Supplementary Methods).

The comprehensive scenario builds directly on the non-federal scenario, layering in federal policies that complement or enhance the policies built out in the Non-Federal scenario. The comprehensive scenario presented here is structured differently than is conventional for assessing national climate action. Rather than assuming that all action is directed from the top-down, the comprehensive scenario reflects a potential alternative approach

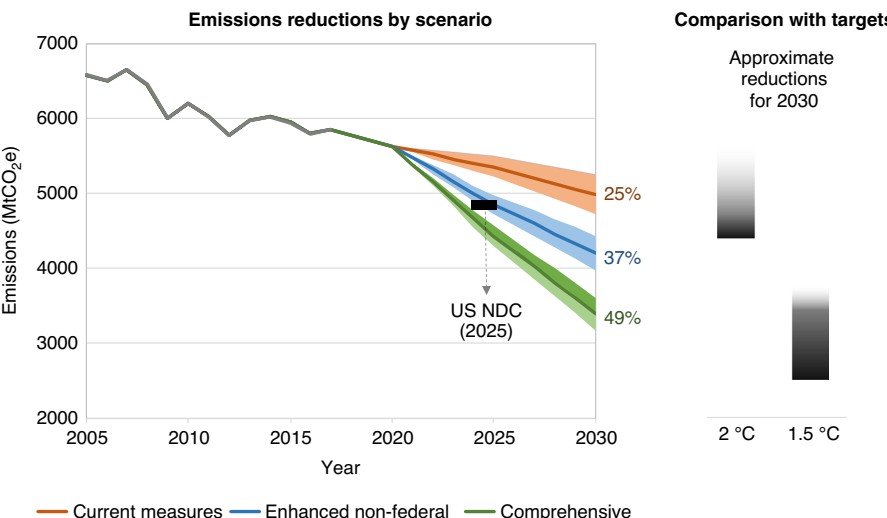

**Fig. 2 Rapidly expanded bottom-up action could reduce U.S. emissions significantly.** Current measures can deliver reductions of 25% below 2005 levels by 2030; enhanced non-federal action can deliver up to 37% reductions, and a comprehensive strategy that layers ambitious federal action on top of state and other sub-national action can deliver up to 49% below 2005 levels by 2030. The gray line shows historical emissions drawn from EPA data. Given large uncertainties and limitations in connecting individual country emissions to global temperature pathways, the shading in the 2 and 1.5 °C bars on the right is indicative.

for generating a national climate strategy—one that is based on early policies that are built and expanded by sub-national actors, and then eventually stitched together and expanded nationally by the federal government. Thus, the comprehensive scenario includes federal policies, where appropriate, that match and build on, rather than simply replacing, the non-federal policies. The federal policies assumed here are a combination of increased regulatory actions (which stem from the executive branch) and new or updated policies from Congress. Such policies include, for example, a package of federal clean energy standards and tax incentives and supporting regulatory actions that complement state efforts and lead to ~50% renewable energy and 75% clean energy nationwide by 2030; and a complete phase out of coal generation by 2030.

We find that current measures would deliver reductions of 19% below 2005 levels by 2025, and 25% below 2005 levels by 2030. Adding in sensitivities, our estimated range for current measures is 17–21% by 2025 and 21–29% by 2030 (Fig. 2). Results for the enhanced non-federal scenario show that, from the base of current commitments, expanding a set of high-impact actions across the Tier 1 and Tier 2 states could reduce U.S. national net emissions from roughly 12% below 2005 levels today, to as much as 37% below 2005 levels by 2030. Finally, the comprehensive scenario, which combines enhanced non-federal action with renewed and ambitious engagement from the federal government, yields reductions of roughly 49% by 2030. Although there is no exact link between a U.S. 2030 emissions level and the global goal of keeping warming under 1.5 °C, the level of reductions under the comprehensive scenario would get the U.S. on a path that is consistent with a global 1.5 °C trajectory. Notable also is that the non-federal scenario—with no additional action or policy changes by the federal government after 2020—could keep the U.S. on a pathway roughly consistent with a global 2 °C trajectory and could set the stage for enhanced action.

Both the enhanced non-federal and comprehensive scenarios deliver an accelerating annual rate of decarbonization: compared to the recent (2005–2017) rate of reductions of 1% per year, we estimate after 2025 the rate accelerates to roughly 2.8% per year in the non-federal and 5.2% in the comprehensive scenario. These rates can be compared to the 4.4% per year average needed to

bring the United States to net zero GHG emissions by mid-century.

To better characterize the range of possible emissions reductions of the scenarios in this study, we examined each scenario under varying assumptions about socio-economic change, technological change, fossil fuel prices, and the size of the land sink. Based on these sensitivities, reductions in the comprehensive scenario could be as low as 46% or as high as 52%. While these are only a subset of uncertainties, they provide an indication of the range of possible emissions associated with different levels of climate action (see "Methods" for further discussion).

Underlying these aggregate figures are some notable results by sector (Fig. 3). The biggest near-term reduction possibilities for which states and cities have significant policy leverage reside in the power sector—primarily through renewable deployment and accelerated coal retirement—and to some degree buildings (through efficiency and electrification policies and building codes) and transportation (via electric vehicles and, in the longer-term, land use planning). However, reductions in line with higher-ambition outcomes are not possible without additional activity in other sectors. In addition, while the largest 2030 reductions arise in the power sector, actions in other sectors are consistent with technological trajectories that would enable deeper emissions reductions beyond 2030.

Changes in each of these sectors are rapid under the highest ambition scenarios, for example, the shifts in electricity generation and capacity build-out by 2030 in the power sector (Fig. 4). New clean energy capacity additions would exceed 75 GW/y, outpacing the recent peak of all source capacity additions of 57 GW in 2003. Transitions of this character are not exclusive to bottom-up approaches and have been documented extensively in previous analyses of deep emissions reductions. While the overall scale of the technological transitions necessary to meet ambitious climate goals remains daunting irrespective of how it is implemented, the central question for the U.S. and other countries is how a bottom-up approach described here might enable these ambitious pathways.

The estimates for the impact of existing policies, including state and city actions, (i.e., the current measures scenario) are broadly

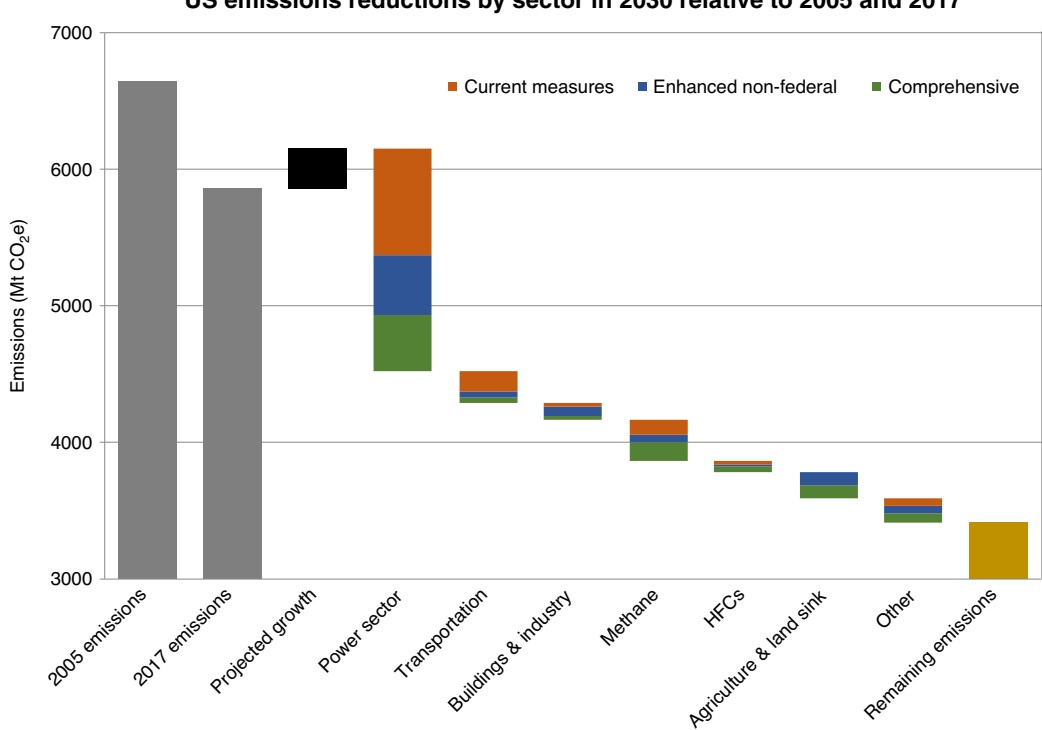

**Fig. 3 Sectoral breakdown of 2030 emission reductions in three scenarios.** Much of the near-term impact is realized through power sector decarbonization, primarily via renewables expansion and rapid coal reduction consistent with recent trends. Many building sector and transportation measures are reflected in the power sector emissions reductions. Note: Scale does not go to zero.

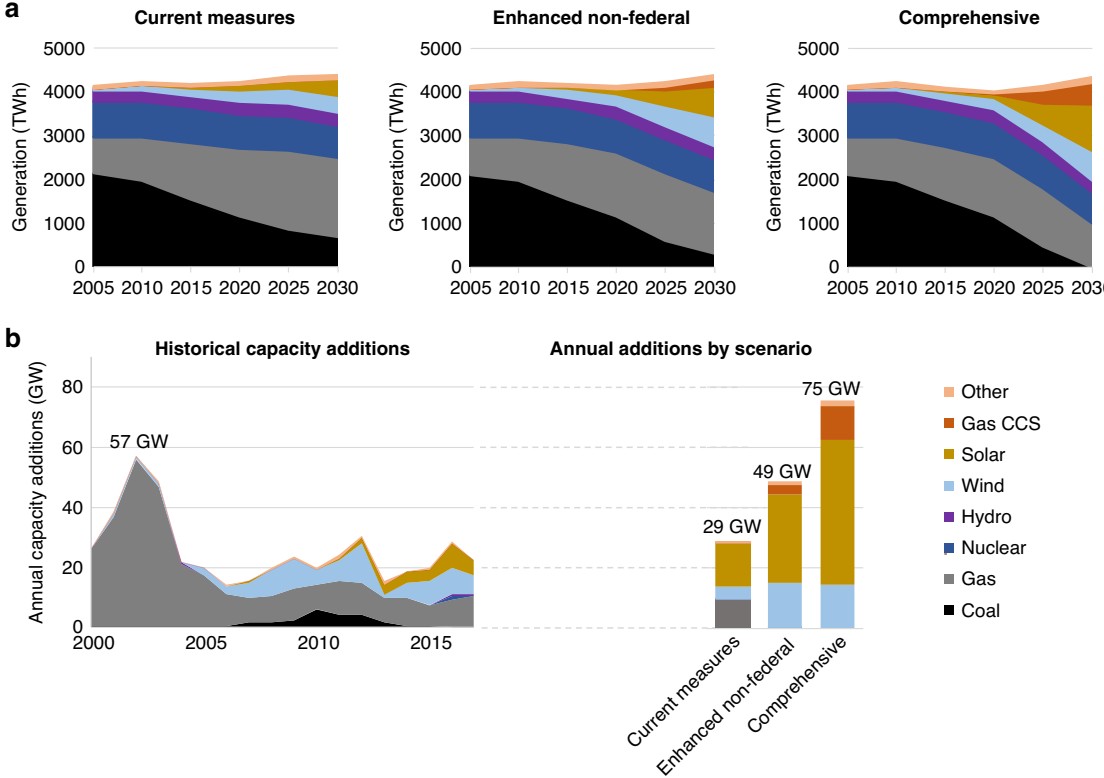

**Fig. 4 The pace of the electricity sector transition between 2020 and 2030 surpasses recent historical experience in the U.S. a** Electricity generation in the current measures, enhanced non-federal, and comprehensive scenarios. **b** Annual power sector capacity additions from 2000 to 2017 and estimated rates of annual additions for the three scenarios.

consistent with coarser estimates that have been made for the potential of U.S. emissions reductions. Kuramochi et al.[30], for example, assessed the impact of current climate policies on 25 key emitters, including the United States. Their global assessment was necessarily less detailed in its roadmap for how climate action could be implemented in the U.S. from sub-national action and is comparable to our current measures projection. Their range of outcomes in 2030 shows a trajectory for the U.S. to achieve roughly a 15–26% reduction, consistent at the upper end with our central estimate of 25%. Similarly, in 2017, the Rhodium Group projected a 15–18% reduction in 2025 and a 14–19% reduction in 2030, relative to 2005[31]. Our current measures results are slightly higher than these, which we attribute primarily to the fact that those estimates do not fully include the sub-national actions that our estimate includes, and they also pre-date much of the recent wave of bottom-up climate commitments described earlier. On the other hand, there is currently no published estimate directly comparable to our higher-ambition enhanced non-federal and comprehensive scenarios.

## Discussion

While it has long been expected that successful climate action will require commitments from national governments and appropriate international coordination[32], the insufficiency of the current global response to climate change clearly points to a need to build more ambitious national actions fed by significant sub-national efforts in organization, coalition building, policy experimentation, and implementation. As countries and their citizens evaluate options for more ambitious strategies for sustainable development and climate-friendly economic opportunities, decarbonization will necessarily be rooted in actions by a diverse range of actors, both governmental and non-governmental. Our analysis shows that such actions have the potential to aggregate to meaningful reductions and to provide a basis to bring emissions in line with global climate goals. The work presented here elucidates in more detail the potential impacts of such a strategy.

In demonstrating the potential for stepped-up action built from sub-national effort, the results suggest an opportunity for enhancing the long-term robustness and effectiveness of climate policy. The processes leading to diversified, non-state policies tie decisions more directly to constituents. As part of a national process, these can help identify new strategies for reducing emissions. In addition, by connecting bottom-up actions to discussions incorporating local or regional benefits, they connect local implementation planning and execution to policies. Such activities can help address the inherent political challenges of generating ambitious national climate policies—and thereby potentially forming a core strategy for an enhanced global response to climate change. On the other hand, such processes are nevertheless subject to political blocking strategies[33], so further elucidation of these mechanisms and their effectiveness is warranted.

Our approach has focused on the case of the United States and has wider applicability and transferability subject to further development and some caveats. We focus on a key subset of actions from states, cities, and businesses. Data limitations on some actions—for example, some business actions—meant that our estimates are likely lower bounds on the magnitude of potential impact from those actions. This data limitation is significant for many actors in the U.S. and is even more acute for large parts of the world, which points to a need for additional data, noting that the currently limited staff capacity and budgets in many organizations may call for a broader global or nationally driven data strategy. Also, our analysis was based on advanced

modeling tools and a large research effort; because of the associated resource needs, a smaller scale strategy that uses a simplified approach could be developed to make such approaches more broadly accessible. Finally, our analysis is focused on emissions impacts, but a related and key question for additional research is to better elucidate the social and governance mechanisms by which bottom-up action can generate deeper and more robust long-term results. This is an area of active research and given the potential centrality to achieving global climate goals will require significant new funding and support.

Thus, as the world looks to increase and sustain climate action over the coming years, catalyzing and empowering sub-national action—though different across countries and contexts—can be essential for all countries. The example of the United States presented here demonstrates how engagement by diverse actors can support higher national action, not only in the conventionally understood sense of implementing goals made by national governments but also helpfully leveraging higher areas of ambition within national boundaries to build the groundwork for more comprehensive and ambitious national policymaking. This is a critical period for all countries to scope new opportunities and to present enhanced nationally determined contributions under the Paris Agreement. As the world grapples with reorganizing quickly around a set of linked sustainable development, economic, and climate goals, forging together sub-national and national action is essential to unlock higher global ambition.

## Methods

**Scenario construction and analytical strategy**. The paper analyzes the potential to reduce greenhouse gas (GHG) emissions in the United States through 2030 under three different scenarios for climate action (see also Supplementary Methods):

- Current measures projects where the U.S. is headed given current policies at national, state, and local levels (Table 1).
- Enhanced non-federal examines outcomes under a broad expansion of cutting-edge climate policies on the part of states, cities, and businesses (Table 2).
- Comprehensive explores a climate strategy integrating aggressive non-federal climate action with renewed federal engagement after 2020 (Table 2).

The analytical approach for modeling each scenario consisted of several steps (Fig. 5 and Supplementary Methods). After developing the scenarios, bottom-up city, state, and business climate policies were aggregated to the state and sector level, eliminating double-counting. For example, we calculated state-level renewable generation resulting from state and city renewable energy policies. Continued technology progress was also considered. The next step was an economy-wide analysis, where we converted the results of the state aggregation into inputs for the U.S.-specific version of the Global Change Assessment Model (GCAM-AP, derived from GCAM-USA) to estimate economy-wide emissions.

A core feature of this two-step analytical approach was the interaction between the state aggregation and economy-wide analysis. Information from GCAM-AP served as an initial representation of activity levels for different sectors. We then aggregated actions in each sector and converted them into sector-appropriate metrics at the state or regional level. These metrics were used as inputs for the economy-wide analysis in GCAM-AP. For a few cases, including coal retirements and the land sector, actions were input directly into GCAM-AP.

**Aggregation of sub-national data and commitments to the state level**. We conducted an analysis of current policies and commitments at multiple scales as well as the potential for accelerated and expanded policies. Sub-national entities implement emissions-related policies for many reasons, including cost savings, consumer benefits, health, economic growth, and climate. For simplicity, in this analysis we refer to any policy that reduces GHG emissions as a climate policy. Our approach to quantifying the impact of city, state, and business actions was informed by existing protocols and methodologies such as the non-state and non-federal action guidance developed through the Initiative for Climate Action Transparency[34], the Global Covenant of Mayors emission scenario methodology[35], and the Greenhouse Gas Protocol mitigation goal standard and policy and action standard[36]. Overall, our process can be summarized as follows:

1. Survey at a minimum all 50 states and the 285 most populous cities in the U.S.
2. Identify a subset of high-impact actions for inclusion in the analysis.
3. Collect the necessary data to quantify each action.

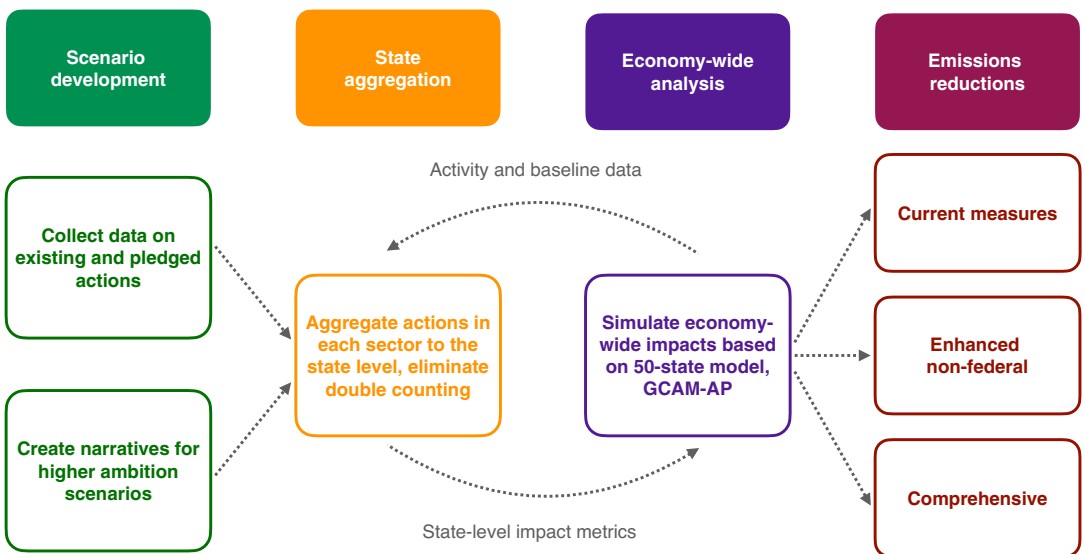

**Fig. 5 Overview of the modeling process.** The economy-wide emissions estimates based on sub-national action are the result of a coupled approach using a state aggregation methodology that is linked to GCAM.

4. Estimate a reference "no policy" scenario for each sector through 2030.
5. Calculate combined impacts for each actor level (e.g., cities and states).
6. Aggregate impacts within each sector to the state level, taking into account overlaps.

**Country level, economy-wide emissions analysis.** The estimates of economy-wide emissions use the results from the state aggregation and GCAM-AP. GCAM-AP is a version of GCAM-USA, which represents the energy and economy components of all 50 states and the District of Colombia. The energy system in GCAM-AP includes detailed representations of depletable primary resources as well as renewable resources such as bioenergy, hydro, solar, wind, and geothermal. GCAM-AP also includes representations of the processes that transform these resources to final energy carriers, such as refining and electric power. The electric power sector includes a range of technologies including those fueled by fossil fuels (with and without CCUS), renewables, bioenergy (with and without CCUS), and nuclear. Future improvements in technology costs and performance are represented exogenously over time. The market equilibrium in each 5-year period is solved by finding a set of market prices such that supplies and demands are equal to one another in all markets. Further details on the GCAM modeling platform can be found on github.com/JGCRI/gcam-core.

We also examine economy-wide results across a range of values for important drivers in our analysis. These include economic growth, population growth, oil and gas prices, renewable energy costs, coal power plant retirements, and the magnitude of the land sink. While these represent a subset of all possible uncertainties, they give a partial indication of the range of possible emissions for each scenario in our analysis. See Supplementary Table 2 for a list of assumptions and sensitivities for the economy-wide analysis.

Further details about the methodological approach developed in this research and additional information on input assumptions can be found in the Supplementary Methods for this article as well as the report *Accelerating America's Pledge: Going All-In to Build a Prosperous, Low-Carbon Economy for the United States* and associated technical appendix[24].

## Data availability
GCAM is an open-source integrated assessment model publicly available at https://github.com/JGCRI/gcam-core. Scenario inputs are summarized in the Supplementary Methods and the source data is available from the corresponding author on reasonable request.

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

## Acknowledgements

This work was conducted as part of the America's Pledge Initiative on climate change, and we are grateful for the sustained input from well over 100 participants in stakeholder meetings with representatives of cities, states, businesses, NGOs, coalitions, research institutions, and other organizations. We are also grateful for the technical comments from Carl Pope, Rick Duke, Angel Hsu, Niklas Höhne, Dan Firger, Jason Mark, Tom Hale, Reed Schuler, Amy Weinfurter, Mark Roelfsema, and Dan Lashof. Elan Strait, Julie Cerqueira, Mariana Panuncio-Feldman, Ryan Finnegan, Lauren Sanchez, David Waskow, Todd Stern, and Mary Nichols, among others, gave helpful input. We thank Todd Edwards, Surabi Menon, and the members of the Climate Action Modeling and Data Analysis (Canada) working group for feedback. Support for this research was provided by the Bloomberg Philanthropies.

## Author contributions

N.E.H., M.R.E., L.C., and R.C. wrote the manuscript. N.E.H., L.C., K.K., H.M., T.C., and K. I. designed the analytical strategy. H.M., T.C., K.I., R.C., M.R.E., A.S., and J. Lou carried out data, aggregation, and modeling in GCAM-AP. T.C., A.C., J. Lund, and K.I. gathered and aggregated data on commitments. P.H., C.F., K.C., H.M. L.C., J. Lund, T.C., K.K., and N.E.H. constructed the policy scenarios with input from all authors; M.R.E., T.C., and H.M. made the figures. C.B., J. Lund, M.I.W., J.J., D.S., W.J., J.W., K.H., J.C.A., M.D., C.H., G.Z., and J.O. provided research on sub-national policies to inform the policy scenarios, data management, and analytical support. N.E.H., P.B., C.F., L.C., K.K, K.C., and M.M. led elements of the overall America's Pledge research program. N.E.H. supervised the study, and all authors contributed to the manuscript or research.

## Competing interests

The authors declare no competing interests.
