## [Peer Review File · Nature Communications]

Reviewers' comments:

Reviewer #1 (Remarks to the Author):

Overall, the piece reads more like a commentary/perspective rather than a research article. I was already familiar with the white paper, "Accelerating America's Pledge: Going All In ...," but I'm not sure that paper, which seems an adaptation of the report, warrants a research article in *Nature Communications*. I do think if the authors shortened the piece and focused on providing some commentary/discussion on the likelihood or factors that might lead to certain scenarios they discuss, it could be an interesting commentary. But there are unfortunately both conceptual and methodological issues that are serious enough to give me pause for its recommendation as a research article.

Conceptual comments:

The title of the piece is "diversified climate action" but the focus on the paper seems to be primarily on U.S. state-level climate action. Much of the introduction discusses a "groundswell" and "bottom-up" participation of "diversified" climate actors, but from what I can tell in reading both the main article and the SI is that the focus is just on U.S. state-level action and re-enactment of federal climate policy. It is unclear at all if the authors included business commitments and how they determined whether there were 'binding mechanisms,' legal or otherwise that would produce the same level of stringency for the various scenarios. The authors should be straightforward and be clear about what this article covers - state-level energy policies and ditch the framing in the introduction about diversification of actors, coalitions, etc. because commitments from these actors do not seem to be evaluated in the models or scenarios. In this context, Fig. 1 is completely misleading as I don't see any evidence in the SI that tribes, healthcare organizations, faith-based organizations' commitments are actually analyzed. There's already a rich literature on state-level climate/energy policies that the authors fail to engage:

Rabe, B. G. (2008). States on steroids: the intergovernmental odyssey of American climate policy. *Review of Policy Research*, 25(2), 105-128.

Peterson, T. D., & Rose, A. Z. (2006). Reducing conflicts between climate policy and energy policy in the US: The important role of the states. *Energy Policy*, 34(5), 619-631.

Goulder, L. H., & Stavins, R. N. (2011). Challenges from state-federal interactions in US climate change policy. *American Economic Review*, 101(3), 253-57.

The authors primarily frame U.S.-state level action as a net-positive, which I think is largely correct. But this is an assumption that merits some further discussion. For instance, the US has also seen some push-back against climate action at the state level (e.g., Ohio's HB 6 law, which subsidizes coal and guts clean energy programs) -- what is the risk of other forms of this kind of push-back based on shifting political leadership? Could state-level action also be weaponized against climate action, if, e.g., there is a 2020 administration committed to ambitious climate action?

Methodological comments:

The authors seem to contradict themselves in the section 'Assessing the current and potential impacts of diversified climate action.' Again, see my comment about the conceptual framing above - the authors should just be straightforward and clear - they are using a state-level policy model to assess national-level scenarios. They seem to dance around some of the challenges of fully incorporating 'bottom-up' action in this section in way that suggests that their model does something (i.e., assess "stakeholder engagement, data collection, aggregation, and modeling" p. 5. Lines 21-23) that other IAMs and GCMs are unable to do, but I don't see any evidence of that.

There is mention of an "Athena" tool but it is not at all described in very much detail in the SI (mentioned briefly in a few paragraphs on p. 2), but it's not clear what "effects of different actions"

(p5., line 29) are included. Why not provide a table in the SI with these details to make it 100% transparent and clear what is actually being evaluated? I found this entire paragraph on page 5 about Athena and its integration into GCAM extremely vague and unclear. Here is some specific language that points to what I mean:

"Explicitly considering interactions to avoid double counting" - ?? what do the authors mean by 'interactions'? It seems they're framing them as negative but they could also be positive (e.g., amplifying, see Chan et al., 2015)?

"Delivers appropriate aggregated inputs back to GCAM-AP" - this statement is also too vague and not reproducible.

"Accounting for overlap" - how? What overlaps? Where? Between actors/sectors/emissions?

"Athena generates estimates of impacts in terms of activity data appropriate to that sector" - is this done on an actor basis, state-level?

A diagram that explicitly makes clear the flows between Athena and GCAM would be 100% helpful here. The authors should remove Fig.1 for being misleading and instead actually provide a diagram of the interaction between Athena and GCAM-AP. Also, Athena has not been published or peer-reviewed, from what I could tell from a web of science search, so I question its rigor or what it's doing, especially because the authors provide scant details on what it includes, how it's doing its "interaction consideration" and "avoiding double counting" and "appropriately aggregating inputs."

Would like to see a clear listing of exactly what policies are included and their estimated emissions reduction impacts. It's unclear from the SI what specific policies are being included and analyzed. In several places, the authors mention "and many other policies," - this is vague and not reproducible.

The authors' approach to measuring Tier 1, Tier 2, and Tier 3 states, based on their likelihood of adopting climate action is a potentially very helpful contribution to attempts to predict the impact of climate action in the US. More details on the criteria that informed their classifications (e.g., how exactly, the outlined criteria for placing a state in each tier were defined) could be useful to other analysts focused on non-state climate action.

The baseline remains unclear - the potential reductions of the current measures, enhanced non-federal measures and comprehensive measures are compared to the NDC target. At times in the SI it seems as if the current measures scenario is used as a kind of baseline, but the implementation of existing subnational commitments is far from certain, and, I would argue, worth understanding on its own terms and for its own contributions from a BAU pathway. The assumptions underlying the trajectory of national emissions in this scenario could be made clearer and more detailed. Page 7 reads: "Figure 2 shows the results of each of the three scenarios. Current Measures would deliver reductions of 25% below 2005 levels by 2030. In contrast, absent any climate policy action, we estimate that absent policies, economic growth would lead to a roughly 4% increase in national emissions in the United States." More details about this baseline scenario (does it project current emissions trends forward based on economic growth? Does it assume current federal policies continue to be implemented?) - in both the text and in Figure 2 would help the reader better understand the potential impact of bottom-up climate action.

How do their assessments compare with other assessments of U.S. "enhanced ambition" or re-engagement of the federal government? Would like to see in the discussion some engagement with other analyses and literature (i.e., the Rhodium Group, Kuramochi et al., 2017 and Kuramochi et al., 2019). No references are made to these other pieces in either the main text or the sensitivity analysis that also analyze future potential emissions reductions from U.S. subnational/non-state actors, which are very similar in their analyses.

Reviewer #2 (Remarks to the Author):

This paper discusses a trending issue to provide the possible mitigation sub-national actors might deliver. The modeling based on GCAM USA and Athena provides reliable quantification of how these sub-national actors can contribute to the climate actions. However, the detail has not been clarified enough for the reader to understand how many firms or cities have been involved in this study, what kind of action has the best effect, etc. The authors raised two interesting questions in the beginning, on 'What bottom-up actor groups could deliver? And how these groups could support greater national actions?' I think more results could be included to discuss these two questions and your findings.

My detailed comments are listed below:

Page 4 line 9: How do you eliminating overlap for the population and GDP? The number seems very impressive that 65% of the US population has been engaged in climate actions, but I'm not very certain if you can detect if one person been engaged in several sub-national coalitions? (I've also roughly go through the report you cited, perhaps I have missed it, but didn't see how they eliminate overlap)

Page 5 line 29: I would suggest you include more detail on the Athena model if this is one of your innovations. The underlying theoretical mechanism needs to be explained here. And I also recommend you to include 'Figure 2 ATHENA Modeling Flow' from your technical appendix of 'Fulfilling America's Pledge' in the supplementary materials.

Page 6: The name of the scenario could be better unified long the text (Bottom-up/All-in vs Enhanced non-federal/Comprehensive).

Page 6 Scenarios: It would be nice to clarify the scenario difference more, either in the explanation of scenario setting or in the supplementary material.

Page 6 line 18: For enhanced non-federal scenario, I'm a bit confused about why you classify the level of potential state ambition and how it works for Athena.

Page 7 Figure 2: How do you address the uncertainty for the three scenarios? Why the emission trajectory (Figure 2) have a range of uncertainty but the sectoral results (Figure 3) don't have?

Page 8 Figure 3: Could be better using transparency to differentiate the historical data and model results for Figure 3 (a).

Page 9 line 30: 'The next year' seems to be quite rough. It would be nice to notice which year it is (as I suppose is this year, 2020, now?).

Page 9 Discussion: You made a very strong argument on why we need to focus on the sub-national actions and I totally agree these were more important than ever for the time being. Yet I would suggest you include more findings from your model to support this idea, with such a big model like GCAM, it would be such a loss to show only such few results.

Response to Referees

NCOMMS-20-08589-A

Diversified climate action is central to rapid decarbonization: The case of the United States

Hultman et al.

Reviewer #1 (Remarks to the Author)

Overall, the piece reads more like a commentary/perspective rather than a research article. I was already familiar with the white paper, “Accelerating America’s Pledge: Going All In ...,” but I’m not sure that paper, which seems an adaptation of the report, warrants a research article in Nature Communications. I do think if the authors shortened the piece and focused on providing some commentary/discussion on the likelihood or factors that might lead to certain scenarios they discuss, it could be an interesting commentary. But there are unfortunately both conceptual and methodological issues that are serious enough to give me pause for its recommendation as a research article.

We thank the reviewer for the many helpful comments and have undertaken to revise the paper in a way that we hope frames our analysis more clearly and also answers the points about the broader contribution. With respect to this specific comment on the tone of the paper, we agreed and have undertaken some restructuring to highlight the research contributions more upfront, and have moved the more broad elements to the discussion section.

Conceptual comments:

The title of the piece is “diversified climate action” but the focus on the paper seems to be primarily on U.S. state-level climate action. Much of the introduction discusses a “groundswell” and “bottom-up” participation of “diversified” climate actors, but from what I can tell in reading both the main article and the SI is that the focus is just on U.S. state-level action and re-enactment of federal climate policy. It is unclear at all if the authors included business commitments and how they determined whether there were ‘binding mechanisms,’ legal or otherwise that would produce the same level of stringency for the various scenarios. The authors should be straightforward and be clear about what this article covers - state-level energy policies and ditch the framing in the introduction about diversification of actors, coalitions, etc. because commitments from these actors do not seem to be evaluated in the models or scenarios.

Thank you for this helpful comment. We absolutely agree that we want to be clear about how we are supporting the thesis with the analysis we have undertaken, so we used these comments as a guide to do that. First, this comment underscores that it is important for us to clarify in the text the extent of our analysis. While it is true that our analysis does include an assessment of state-level energy policies, it does go much deeper than this, and this is one reason we feel that it is a helpful and new contribution to the literature. In fact, we have assessed a large number of diverse city, state, and business policies, covering not just energy generation but also transportation, buildings, the land sector, methane, HFCs, and others. A significant data and analytical effort went into assessing how these diverse and overlapping policies would impact overall U.S. emissions. Accordingly, in the text, we have sought to clarify in all instances what we are doing with respect to our analysis.

As one example, we have modified our abstract to provide a clear statement of what work we did: “We aggregated U.S. state, city, and business commitments within a global integrated assessment model (GCAM), to assess how a comprehensive national climate strategy can be built from expanded actions at subnational levels.” We also added the following statement at the end of our introduction: “In this paper, we describe how a significant subset of those actors in the United States—including states and cities, and some businesses—are implementing policies that significantly reduce economy-wide emissions, and how expanded action from those actors can provide a higher platform for potential additional Federal action.”

In doing this we believe there is one contribution of our analysis that we should clarify for the reviewer, and one area that we agree is a legitimate limitation of our analysis.

The first area that we believe was a matter of misunderstanding is that, while we assess all of these diverse actions from cities and states, our methodology does have us aggregate these actions to the state level for purposes of undertaking an economy-wide assessment of emissions impacts. In other words, we run a 50-state model in which each state's emissions are driven by a combination of both state-wide, and city policies. As such, while our analysis ultimately is based on a 50-state assessment, the drivers of action are carefully assessed. This is the function of the "Athena" model that we describe in the paper and the SI—and which we have undertaken significant revisions in the text to clarify. Athena is the structure within which we carefully eliminate the potential double counting that can arise from having, for example, a city policy that might be partially or fully subsumed by a similar state policy. In light of the reviewer's comments, we have clarified the language about our methodology, both in the main text and in the methodology section and SI.

In another way, the reviewer is absolutely correct. That is, we do not seek to incorporate actions of entities that are not states, cities, or businesses. For example, the "We Are Still In" coalition includes a number of specific categories of groups, such as tribal groups and universities, that have relatively few policy levers and a relatively small potential impact on emissions. We do not attempt to quantify the impacts of those types of actors. There are two reasons for this. First, there is not usually sufficient baseline and other data for us to make a confident assessment. Second, it is also our view that compared to the city and state policies, the absolute impact of such activities would be small. Business action is one category that could potentially impact the results, although likely in a smaller way, and we can only partially cover that area. Unfortunately, most business reporting is done on a company-wide rather than a plant-by-plant basis, so the inability to pinpoint geographical location of reductions precludes a confident assessment of many business actions. However we do evaluate a limited number of actions, which are listed in the summary table in the Methods and elaborated in more detail in the SI. These include 24 utility decarbonization goals, voluntary corporate actions through EPA Gas STAR, and voluntary HFC action through GreenChill. We evaluate these as so-called *pledged actions* and therefore do not include them in our Current Measures scenario. We do include these actions in the more ambitious scenarios, although they are frequently overridden by other, more ambitious assumptions.

We thus view our assessment as a lower bound on the impact of subnational actions. At the same time we think that given other uncertainties, our numbers are robust. Because we are capturing what we believe are the most significant actions, we believe that the analysis does effectively illuminate the potential impacts of subnational action on the U.S. national emissions trajectory.

In this context, Fig. 1 is completely misleading as I don't see any evidence in the SI that tribes, healthcare organizations, faith-based organizations' commitments are actually analyzed.

Thank you for the comment. In this introductory position in the paper, we are focusing primarily on simply conveying the idea—which we think may not be obvious to people not deeply engaged in this field—that there is a broad diversity of action in the U.S. To this end, we believe that the map helps to illustrate a broader concept, which is the breadth of action across the United States, and thus we have retained this figure to help support the motivation for better understanding the diversity of this action. It was placed in the first section that highlights the overall trend of sub-national action and was meant to give the reader a sense of scale. At the same time, we fully agree with the reviewer that we don't want to mislead readers into thinking we analyzed every action by these coalitions. The dots on the figure do represent all actors, and as the reviewer points out correctly, some of these we did not explicitly model. As such we have clarified the purpose of the figure in the manuscript and figure caption. We have added the following clarifications:

Despite the scale of this groundswell of diverse new climate commitments, the central question remains how such coalitions, acting outside national government authorities, can impact overall national emissions trajectories—and, by extension, how broader action across countries can affect the global emissions trajectory. **The research presented here does this by estimating the impact of a subset of the diverse**

actors shown in Figure 1: the U.S. states and 285 largest cities who have the most impact on overall U.S. emissions.

And in the caption to Figure 1:

a) States, cities, businesses, universities, faith-based and other organizations now number greater than 3800. **This study assessed the most significant subset of these actors, focusing on states and the 285 largest cities.**

In evaluating the 50 states and 285 largest cities, we found 168 cities with policies that we were able to include. We describe this process in the text, the Methods section, and in the Supplementary Information. To be fully clear, we have also created a new figure, which is Figure 2 in the Supplementary Information, to show the geographic locations of the cities included.

There's already a rich literature on state-level climate/energy policies that the authors fail to engage:

Rabe, B. G. (2008). States on steroids: the intergovernmental odyssey of American climate policy. *Review of Policy Research*, 25(2), 105-128.

Peterson, T. D., & Rose, A. Z. (2006). Reducing conflicts between climate policy and energy policy in the US: The important role of the states. *Energy Policy*, 34(5), 619-631.

Goulder, L. H., & Stavins, R. N. (2011). Challenges from state-federal interactions in US climate change policy. *American Economic Review*, 101(3), 253-57.

Thank you for these suggestions. We agree that the literature on state and non-state climate action goes back to important earlier works, and we absolutely want to converse with that earlier literature. In our previous draft, we did refer back briefly to some of this literature; for example, the Rabe 2008 paper was already cited and discussed in that version. Nevertheless, the reviewer's point is well taken. Accordingly, we have expanded the section referring to this literature more to more fully reflect how those ideas relate to our own work. As part of this, we have also integrated discussion of the additional two papers suggested by the reviewer.

We have also enhanced our discussion of the unique character of the particular circumstances in the United States today, with increased international engagement on climate change, severely diminished federal leadership, and an unprecedented level of explicit, bottom-up action on climate in the United States. This circumstance brings into sharper focus and raises the relevance more than ever before of the role that bottom-up actors might play in climate mitigation. It is this emerging circumstance that provides the motivation for this paper.

The authors primarily frame U.S.-state level action as a net-positive, which I think is largely correct. But this is an assumption that merits some further discussion. For instance, the US has also seen some push-back against climate action at the state level (e.g., Ohio's HB 6 law, which subsidizes coal and guts clean energy programs) -- what is the risk of other forms of this kind of push-back based on shifting political leadership? Could state-level action also be weaponized against climate action, if, e.g., there is a 2020 administration committed to ambitious climate action?

This is an incisive point that raises some important political questions, and one that we agree with fully. We have added some text to reflect this issue in the paper, specifically:

It is important to note that the while the aggregate effects of sub-national action may serve to build a higher overall level of action in the United States, individual sub-national actions are not always consistent with long-term climate goals or clean energy transitions; they may also lead to conflicts and economic inefficiencies. States have sometimes also opposed federal climate policies through court actions.

Methodological comments:

The authors seem to contradict themselves in the section ‘Assessing the current and potential impacts of diversified climate action.’ Again, see my comment about the conceptual framing above - the authors should just be straightforward and clear - they are using a state-level policy model to assess national-level scenarios. They seem to dance around some of the challenges of fully incorporating ‘bottom-up’ action in this section in way that suggests that their model does something (i.e., assess “stakeholder engagement, data collection, aggregation, and modeling” p. 5. Lines 21-23) that other IAMs and GCMs are unable to do, but I don’t see any evidence of that.

Thank you for the comment. We addressed the bigger picture framing question above, as well as how our approach does provide a deeper assessment of bottom up action than previous work. There is a second dimension to this comment here that we can respond to here. Our model is really in three parts, and there is a fourth element that the reviewer mentions here. First, we do have a data collection component of the work, which is both challenging but is an essential part of the contribution of this paper. We obtained data on subnational goals, policies, baselines, and targets from a number of sources, including CDP, Under2MOU, C2ES, ICLEI, carbon n , ACEEE, Second Nature, BEA, U.S. EPA, U.S. DOE (including the SLED tool), and others. Data sources are cited in our sector-level descriptions in the Supporting Information. Second, we then place these inputs in a context of a broader state, national, and global baseline, and then carefully aggregate the impacts on that baseline up to the state level. That is the function of our Athena tool, which is calibrated to an overall GCAM baseline. Third, we then take these state level inputs (i.e. the impacts of subnational policies at the state level) and run them in GCAM to account for economy-wide interactions across the national and global climate, energy, and land systems. That three part process is novel for IAMs.

The other dimension mentioned by the reviewer is stakeholder engagement. This did not impact the modeling strategy above, but instead was used to establish our scenarios. Rather than just make assumptions for subnational action, our process engaged with actors to see what they found to be the most plausible policies to expand at the subnational level. We therefore think that adds an important dimension for the integrated roadmap strategy that is exemplified by this research. However, we do not have space to really describe this process in detail for this paper, so we just mention it briefly. It might be valuable to do as a separate white paper or publication.

In response to these helpful comments, we have added additional material on our overall modeling process and the Athena model in particular, as follows:

P.4 - Clarification of methods

To address these challenges, we developed a new, comprehensive strategy to assess the scope and **emissions impact of non-federal climate actions in the United States. We do this by integrating data on baselines and sector-specific policies and commitments into an economy-wide estimate of the U.S. emissions trajectory to 2030. The overall process involved steps of data collection, baseline development, aggregation, and modeling, with parallel stakeholder engagement to help develop the policy platform for scenarios (Described further in the Methodology section and in the SI).** The resulting approach combines detailed accounting of bottom-up actions with an integrated assessment model to understand their overall emission impacts.

There is mention of an “Athena” tool but it is not at all described in very much detail in the SI (mentioned briefly in a few paragraphs on p. 2), but it’s not clear what “effects of different actions” (p5., line 29) are included. Why not provide a table in the SI with these details to make it 100% transparent and clear what is actually being evaluated? I found this entire paragraph on page 5 about Athena and it’s integration into GCAM extremely vague and unclear.

Thank you for the suggestion. We have added additional material in the SI to clarify what Athena is and how it works, as well as how it interacts with GCAM. In addition, we agree that the goals and purpose of Athena needed to be made more explicit in the main text. Accordingly, we added the following on p. 5:

In assessing actions that impact emissions with overlapping or nested geographical coverage—for example, a city policy on renewables within a state that has a renewables target— there is a need to eliminate double-counting and otherwise account for potential interactions in the areas of overlap. At the same time, it is also important to ensure that baselines and input data across jurisdictions are consistent to ensure that the resulting broader projection is robust. To address these issues, we developed an aggregation methodology (which we call Athena) that draws from a well-established set of inputs from GCAM and then aggregates the effects of diverse climate policies while explicitly considering interactions and avoids double counting within each sector.

Here is some specific language that points to what I mean:

“Explicitly considering interactions to avoid double counting” - ?? what do the authors mean by ‘interactions’? It seems they’re framing them as negative but they could also be positive (e.g., amplifying, see Chan et al., 2015)?

We address potential issues with double counting in two ways. First, we account for overlap between policies within a sector in Athena (for example, cities within a state that both have a RPS). Second, we model interactions between sectors in GCAM (for example, impacts of electric vehicle policies on electricity demand). We have added further discussion of our process for accounting for overlap in the SI (see *Avoiding Double Counting*). To avoid confusion, we changed “interactions” to “overlap” in the sentence referenced by the reviewer.

We recognize that in practice policies can be overlapping or reinforcing in terms of impact. We discuss how we account for overlap on a sector-by-sector basis in more detail in the *Sector Descriptions* section of the SI. For example, for RPS policies:

We aggregated renewable energy generation resulting from the state, city, and utility actions described above at the state level, partially accounting for overlap across the three different levels of action. We accounted for city policies using a net percentage approach, where additional renewable demand from city goals in percentage terms (beyond the percentage required by state policies) is added to the state total. For example, a city with a 50% goal in a state with a 40% RPS add a net 10% renewable generation to its electricity load. Alternatively, if the city’s projected renewable target were less than the state’s 40% target, no additional renewable demand would be included. We applied the same net percentage approach when accounting for utility targets.

“Delivers appropriate aggregated inputs back to GCAM-AP” - this statement is also too vague and not reproducible.

We agree this statement is vague, and was also unnecessary. We have deleted it.

“Accounting for overlap” - how? What overlaps? Where? Between actors/sectors/emissions?

Here we meant between actors -- for example, a city within a state that both have a renewable portfolio standard. We have clarified this in the text.

“Athena generates estimates of impacts in terms of activity data appropriate to that sector” - is this done on an actor basis, state-level?

This is done on a sector level. We clarified this.

A diagram that explicitly makes clear the flows between Athena and GCAM would be 100% helpful here. The authors should remove Fig.1 for being misleading and instead actually provide a diagram of the interaction between Athena and GCAM-AP. Also, Athena has not been published or peer-reviewed, from what I could tell from a web of science search, so I question it's rigor or what it's doing, especially because the authors provide scant details on what it includes, how it's doing its “interaction consideration” and “avoiding double counting” and “appropriately aggregating inputs.”

Thank you for this suggestion. We have created a new diagram illustrating the modeling process and feedbacks (in the Methods section of the main paper).

Would like to see a clear listing of exactly what policies are included and their estimated emissions reduction impacts. It's unclear from the SI what specific policies are being included and analyzed. In several places, the authors mention “and many other policies,” - this is vague and not reproducible.

We have now added a table listing the policies included in each scenario (in the Methods section of the main paper). We describe these policies in more detail in the *Sector Descriptions* section of the SI. Since we model emissions reductions across the U.S., we do not estimate emissions reductions resulting from individual policies. However, we have included a new figure in the main text that breaks down emissions reductions contributions by sector.

The authors' approach to measuring Tier 1, Tier 2, and Tier 3 states, based on their likelihood of adopting climate action is a potentially very helpful contribution to attempts to predict the impact of climate action in the US. More details on the criteria that informed their classifications (e.g., how exactly, the outlined criteria for placing a state in each tier were defined) could be useful to other analysts focused on non-state climate action.

Thank you for this suggestions, we have added a listing of the criteria in the main text:

In parallel, we generated three groupings of states that reflect the likelihood of accelerated action, based on attributes such as membership in climate organizations, vocal leadership in support of climate action, ambitious emissions reduction targets or standards, and on-the-books climate poli-cies. In our scenario, Tier 1 states realize levels of ambition roughly equivalent to today's leading states like California, Tier 2 states lag but still move toward the level of ambition of Tier 1, and Tier 3 states which do not implement major changes (SI).

The baseline remains unclear - the potential reductions of the current measures, enhanced non-federal measures and comprehensive measures are compared to the NDC target. At times in the SI it seems as if the current measures scenario is used as a kind of baseline, but the implementation of existing subnational commitments is far from certain, and, I would argue, worth understanding on its own terms and for its own contributions from a BAU pathway. The assumptions underlying the trajectory of national emissions in this scenario could be made clearer and more detailed. Page 7 reads: “Figure 2 shows the results of each of the three scenarios. Current Measures would deliver reductions of 25% below 2005 levels by 2030. In contrast, absent any climate policy action, we estimate that

absent policies, economic growth would lead to a roughly 4% increase in national emissions in the United States.” More details about this baseline scenario (does it project current emissions trends forward based on economic growth? Does it assume current federal policies continue to be implemented?) - in both the text and in Figure 2 would help the reader better understand the potential impact of bottom-up climate action.

We have added a new section to the SI to describe our reference scenario in further detail:

Initial data from GCAM-AP was generally interpreted in Athena as a no-policy, reference scenario in which subnational policies, and some key federal policies, are not represented. Thus, the full impact of policies was applied to the baseline projections without needing to address overlap. Exceptions to this assumption and cases where any subnational policies were already embedded in the baseline are discussed in the sector descriptions in Section 5.

The GCAM-AP reference case scenario does include certain federal-level policies that have significant impacts within the sectors modeled. These include the federal production tax credit (PTC) and investment tax credit (ITC) in the renewable energy sector and federal fuel economy standards in the transportation sector. While sectoral modeling results in Athena typically represent the impact of state, city, or business policies only, final modeling results from GCAM-AP account for the combined impacts of these federal-level policies and the non-federal impacts from Athena. In addition, two types of federal policies not already included in GCAM-AP were explicitly modeled in Athena and aggregated with non-federal actions before being passed back to GCAM-AP as inputs. These were the U.S. EPA Section 608 refrigerant management policy for HFCs and regulations to reduce fugitive emissions in the oil and gas sector.

How do their assessments compare with other assessments of U.S. “enhanced ambition” or re-engagement of the federal government? Would like to see in the discussion some engagement with other analyses and literature (i.e., the Rhodium Group, Kuramochi et al., 2017 and Kuramochi et al., 2019). No references are made to these other pieces in either the main text or the sensitivity analysis that also analyze future potential emissions reductions from U.S. subnational/non-state actors, which are very similar in their analyses.

There have been a few other assessments of the impact of subnational action on emissions both at the global level and, in a couple of cases, in the United States specifically. We discussed the global perspectives in our framing sections, but the reviewer makes a good point about two analyses that look specifically at U.S. emissions under current measures. Kuramochi and others are co-authors on the two studies that we noted in the first draft, and we have now added explicit reference paper suggested by the reviewer as well as one new one. Substantively, this group is strong and we agree they have done good work looking at global impacts in particular. Because their focus is global, their assessment of the U.S. is necessarily less detailed. Their numbers are roughly in line with ours, but their aggregate approach does not allow for the explicit linking of policy roadmaps that our more detailed analysis allows. In this sense, it remains our view that the methodological approach we present here is an important advancement. Also, our are broadly consistent with Rhodium Group’s study from 2017, although their lack of published methodologies makes a detailed comparison of comparability between their results on ours somewhat challenging. We have added a reference to this in our paper, because we agree with the reviewer that some comment is helpful here. We have added the following to the text:

There is currently no published estimate directly comparable to our higher-ambition *Enhanced Non-Federal* and *Comprehensive* scenarios. On the other hand, the estimates for the impact of existing policies, including state and city actions (i.e., the *Current Measures* scenario), are broadly consistent with coarser estimates that have been made for the potential of U.S. emissions reductions. Kuramochi et al., for example, assessed the impact of current climate policies on 25 key emitters, including the United States. Their global assessment was necessarily less detailed in its roadmap for how climate action could be implemented in the U.S. from sub-national action, and the best analogue to our assessment is to our *Current*

Measures projection. Their range of outcomes in 2030 shows a trajectory for the U.S. to achieve roughly a 15-26% reduction, consistent at the upper end with our central estimate of 25%. Similarly, in 2017, the Rhodium Group used publicly available versions of GCAM and NEMS to forecast near-term U.S. emissions, and they projected a 15-18% reduction in 2025 and a 14-19% reduction in 2030, relative to 2005. Our Current Measures results are slightly higher than these, which we attribute to the fact that those estimates do not fully include the sub-national actions that our estimate includes, and they also pre-date much of the recent wave of bottom-up climate commitments described earlier

Reviewer #2 (Remarks to the Author):

This paper discusses a trending issue to provide the possible mitigation sub-national actors might deliver. The modeling based on GCAM USA and Athena provides reliable quantification of how these sub-national actors can contribute to the climate actions.

Thank you for the comment and reflection on the relevance and reliability of our analysis.

However, the detail has not been clarified enough for the reader to understand how many firms or cities have been involved in this study, what kind of action has the best effect, etc.

We have provided more detail about this now in the paper, and have enhanced our descriptions of this wherever possible.

The authors raised two interesting questions in the beginning, on ‘What bottom-up actor groups could deliver? And how these groups could support greater national actions?’ I think more results could be included to discuss these two questions and your findings.

Thank you for this comment. We have added some additional results into the main text, including a new figure, to provide additional context for and support of our argument.

My detailed comments are listed below

Page 4 line 9: How do you eliminating overlap for the population and GDP? The number seems very impressive that 65% of the US population has been engaged in climate actions, but I’m not very certain if you can detect if one person been engaged in several sub-national coalitions? (I’ve also roughly go through the report you cited, perhaps I have missed it, but didn’t see how they eliminate overlap)

Here we are calculating the percent of the U.S. population whose subnational governments have declared support for the Paris Agreement. We avoid double counting by not counting the population of any cities within a state that has also declared support. We have added further clarification in the text.

Page 5 line 29: I would suggest you include more detail on the Athena model if this is one of your innovations. The underlying theoretical mechanism needs to be explained here. And I also recommend you to include ‘Figure 2 ATHENA Modeling Flow’ from your technical appendix of ‘Fulfilling America's Pledge’ in the supplementary materials.

We have added a paragraph in the main text describing Athena, and have improved and included the figure suggested (Figure 2 Athena modeling flow). It is now in the Methodology section of the main text.

Page 6: The name of the scenario could be better unified long the text (Bottom-up/All-in vs Enhanced non-federal/Comprehensive).

Thank you, we have updated all of our scenario names to be consistent.

Page 6 Scenarios: It would be nice to clarify the scenario difference more, either in the explanation of scenario setting or in the supplementary material.

We have added substantial elaboration on the scenarios in the main text Methods and in the SI. We describe the conceptual differences in the main text and in bullets in the Methods:

- *Current Measures projects where the U.S. is headed given current policies at national, state, and local levels (Table 1).*
- *Enhanced Non-Federal examines outcomes under a broad expansion of cutting-edge climate policies on the part of states, cities, and businesses (Table 2).*
- *Comprehensive explores a climate strategy integrating aggressive non-federal climate action with renewed federal engagement after 2020 (Table 2).*

Table 1 and 2 provide descriptions of the specific actions included in our analysis. We also included more descriptions of our methods in the new *Scenario Development* section of the SI as well as in the *Scenario Descriptions*.

Page 6 line 18: For enhanced non-federal scenario, I'm a bit confused about why you classify the level of potential state ambition and how it works for Athena.

We group states in different tiers to reflect their willingness to engage in climate action. We identified these states by attributes such as membership in climate organizations, vocal leadership in support of climate action, ambitious emissions reduction targets or standards, and on-the-books climate policies. We added more detail on this in the main text when we describe the *Enhanced Non-Federal* scenario.

Page 7 Figure 2: How do you address the uncertainty for the three scenarios? Why the emission trajectory (Figure 2) have a range of uncertainty but the sectoral results (Figure 3) don't have?

We added further discussion of the treatment of sensitivities in the main text:

To better characterize the range of possible emissions reductions of the scenarios in this study, we examined each scenario under varying assumptions about socioeconomic change, technological change, fossil fuel prices, and the size of the land sink. Based on these sensitivities, reductions in the Comprehensive scenario could be as low as 46% or as high as 52%. While these are only a subset of uncertainties, they provide a range of possible emissions associated with different levels of climate action. (See Methods for further discussion.)

We also add discussion in the Methods:

We also examine economy-wide results across a range of values for important drivers in our analysis. These include economic growth, population growth, oil and gas prices, renewable energy costs, coal power plant retirements, and the magnitude of the land sink. While these represent a subset of all possible uncertainties, they give a partial indication of the range of possible emissions for each scenario in our analysis. See Table 2 in the supporting information for a list of assumptions and sensitivities for the economy-wide analysis.

We further discuss these in the SI. For visual simplicity, we include our sensitivities in our line graph showing emissions reductions but not in the sectoral results.

Page 8 Figure 3: Could be better using transparency to differentiate the historical data and model results for Figure 3 (a).

We added clarification in the caption that the grey line indicates historical data.

Page 9 line 30: ‘The next year’ seems to be quite rough. It would be nice to notice which year it is (as I suppose is this year, 2020, now?).

We have changed the text to “This is a critical period” - given that now the timeline for submitting new NDCs may spill into next year, but no decisions have been made yet, we thought it best not to anchor to a specific year.

Page 9 Discussion: You made a very strong argument on why we need to focus on the sub-national actions and I totally agree these were more important than ever for the time being. Yet I would suggest you include more findings from your model to support this idea, with such a big model like GCAM, it would be such a loss to show only such few results.

Thank you for the comment. The reviewer is correct that GCAM produces a vast quantity of outputs, many of which are interesting and worth discussing. So we are happy to have the invitation to share more results. Mindful that we still have a space constraint, we have added one additional figure and some associated text: “we have also prepared the sectoral policy numbers by state that serves as a building block to the state-level modeling. These files are attached for the review, and will be made available upon request (due to the inclusion of proprietary data).”

This figure shows a sectoral breakdown of where the emissions reductions come from within our model framework. Combined with the deeper dive on the power sector, we think this gives a more complete picture of our results.

REVIEWER COMMENTS

Reviewer #1 (Remarks to the Author):

Response to Referees NCOMMS-20-08589-A Diversified climate action is central to rapid decarbonization: The case of the United States Hultman et al.

Reviewer #1 (Remarks to the Author)

Overall, the piece reads more like a commentary/perspective rather than a research article. I was already familiar with the white paper, "Accelerating America's Pledge: Going All In ...," but I'm not sure that paper, which seems an adaptation of the report, warrants a research article in Nature Communications. I do think if the authors shortened the piece and focused on providing some commentary/discussion on the likelihood or factors that might lead to certain scenarios they discuss, it could be an interesting commentary. But there are unfortunately both conceptual and methodological issues that are serious enough to give me pause for its recommendation as a research article.

We thank the reviewer for the many helpful comments and have undertaken to revise the paper in a way that we hope frames our analysis more clearly and also answers the points about the broader contribution. With respect to this specific comment on the tone of the paper, we agreed and have undertaken some restructuring to highlight the research contributions more upfront, and have moved the more broad elements to the discussion section.

Response:

- While the authors have provided more context for city and state-level climate policy in the U.S., there is still a major gap in the conceptual framing of the paper that remains unresolved. The title of the paper remains "diversified climate action," which the authors describe in the introduction to mean diverse coalitions and groups of subnational and non-state actors. While this topic is of timely interest given the lack of national U.S. leadership on climate change and energy, the paper still remains primarily focused on state-level U.S. climate action, mandatory or binding policies (in the authors' own words), and broad, sector-level actions that are modeled at a highly-aggregated, sectoral level for a picture of U.S. emissions under three scenarios of varying ambition levels. The primary question I kept asking myself in re-reading this paper and the SI is the question of how much of the policies evaluated (i.e., renewable portfolio standards, energy efficiency standards, coal phase-outs) really reflect a "new diversity" of "bottom-up actor groups" as opposed to what states have been doing all along. In the U.S. context, climate policy has always been driven primarily by states - California is a prime example and the policies included in the 'Current Measures' scenario are reflective of states' leadership on climate policy. In fact, the U.S. has never had a comprehensive, nationwide climate policy. The authors identify key gaps with respect to bottom-up climate actions on p. 2 lines 32-34 that there is limited understanding of what bottom-up actor groups could deliver and how they could support greater national actions. They also claim on p. 3 lines 5-7 that the paper "describe[s] how a significant subset of those actors, including states and cities, and some businesses are implementing policies that significantly reduce economy-wide emissions." I unfortunately still

feel that while the paper has much potential to accomplish these goals, the article as it currently stands fails to achieve these goals. The authors do not answer the 'how' part (i.e., it would be fascinating if the authors could disentangle which policies, particularly if implemented at the business or local government level, could actually build the meaningful catalytic chain of policies that would "support greater national actions", similar to what Martin and Saikawa (see the reference below) do in their piece examining state-level policies' effectiveness at driving down power sector emissions). In terms of the 'what' part of bottom-up groups contributions, the greater detail provided in the SI still appears to be primarily focused on state-level policies. It would be helpful to understand the magnitudes of these contributions from various policies and actor groups so that a policymaker could understand the relative impacts of different groups and policies. I'm reminded of Meckling et al. (2015)'s piece on winning coalitions for climate and energy policy - a detailed analysis of how different actors engage to catalyze policies at the local levels and within the private sector to generate support for national-level policy could be a more compelling framing. But this article still primarily focuses on highly-aggregated, sector level contributions and doesn't go into this level of granular detail.

- I'm not saying that this article doesn't have value. It does, but it's still not framed appropriately and as such, it falls short of delivering on its current promise. If the authors would simply reframe the article as an investigation of state-level climate and energy policies' impacts and contributions in a current measures scenario, and then what these impacts could be if more states (or if there existed a national policy that applied to and was implemented by all states) adopted and implemented them, then this article to me would be a lot more conceptually coherent in the context of the very valid research question of how the U.S. can continue making progress on climate change in the absence of coordinated U.S. action. Even more meaningfully, this type of analysis could then allow for policymakers to envision the types of reductions possible with nationwide, coordinated U.S. action. In many contexts outside of the U.S., for example, we can see how national-level policies have been effective in achieving climate goals (i.e., China, Costa Rica, among others). The "bottom-up" part of state-driven climate policy in the U.S. represents a failure of national policy in some sense.
- Here are a few specific lines that illustrate what I mean by problematic framing of the paper:

P. 5 Lines 15-18: "Despite the scale of this groundswell of diverse new climate commitments, the central questions remains how such coalitions, acting outside national government authorities, can impact overall national emissions trajectories—and, by extension, how broader action across countries can affect the global emissions trajectory." Characterizing predominantly state-level climate policy doesn't answer the question about how "diverse new climate commitments" and how "such coalitions, acting outside national government authorities" can impact overall trajectories. The US's climate trajectory has always been impacted by states, so I'm not sure what else is being added that is new or helps shed light on this question.

P. 3 Line 18-19: "The United States today provides a globally critical testing ground for essential questions on how subnational commitments can contribute to better outcomes for overall national climate strategies." - ok but major caveat that the U.S. context is really specific to the federated U.S. system - there are limits to the applicability. The authors should discuss this limitation.

P. 3 Lines 28-29: "The evolution of state policies next to Federal policies in the United States, has raised the prospect of national climate policies driven by enhanced "bottom up" action." Again, U.S. climate

policy has always been driven by states.

P. 6 Lines 33-36: “ In order to keep modeling conservative and analytically rigorous, the scenario did not include commitments to reduce GHG emissions on their own unless backed by a binding mechanism (e.g. the Regional Greenhouse Gas Initiative (RGGI) and California’s cap). A summary of these policies is provided in Table 1 and further details are in the SI.” These are again, mandatory, legally-backed policies, which are distinct from the largely voluntary nature of private sector, bottom-up action. This would not be problematic if the paper’s framing were recast as an examination of state-level climate policy and future contributions.

P.12 Lines 10-13: “The work presented here elucidates in more detail a mechanism for such interactions, providing opportunities for diverse actors to generate their own goals and commitments as part of a national process and to better understand their contributions to the overall emissions out-comes.” The mechanisms are not clear - again, the policies and actions are aggregated to the state-level so it’s not clear what mechanisms the authors are referring to.

- My second major concern is still about the Methods. While the authors have provided much more information regarding the policies evaluated in the Tables and SI that have been added, I’m concerned that the method is not reproducible given the insufficient details provided. For a scientific paper to be published as an article in a high-profile journal like Nature Comms, the method needs to be reproducible. The major part that is still not clear to me is the Athena component. I now understand it much better (see below comment), but it appears to be a decision-making process, rather than a model or tool, to determine which non-Federal actors’ commitments are included or are considered overlapping with state efforts. However, insufficient details are given that would allow for reproducibility of their analysis.

References:

- Martin, G., & Saikawa, E. (2017). Effectiveness of state climate and energy policies in reducing power-sector CO 2 emissions. *Nature Climate Change*, 7(12), 912-919.
- Meckling, Jonas, Nina Kelsey, Eric Biber, and John Zysman. "Winning coalitions for climate policy." *Science* 349, no. 6253 (2015): 1170-1171.

Other responses to comments written in green below.

Conceptual comments: The title of the piece is “diversified climate action” but the focus on the paper seems to be primarily on U.S. state- level climate action. Much of the introduction discusses a “groundswell” and “bottom-up” participation of “diversified” climate actors, but from what I can tell in reading both the main article and the SI is that the focus is just on U.S. state-level action and re-enactment of federal climate policy. It is unclear at all if the authors included business commitments and how they determined whether there were ‘binding mechanisms,’ legal or otherwise that would produce the same level of stringency for the various scenarios. The authors should be straightforward and be clear about what this article covers - state-level energy policies and ditch the framing in the introduction about diversification of actors, coalitions, etc. because commitments from these actors do not seem to be evaluated in the models or scenarios.

Thank you for this helpful comment. We absolutely agree that we want to be clear about how we are supporting the thesis with the analysis we have undertaken, so we used these comments as a guide to do that. First, this comment underscores that it is important for us to clarify in the text the extent of our analysis. While it is true that our analysis does include an assessment of state-level energy policies, it does go much deeper than this, and this is one reason we feel that it is a helpful and new contribution to the literature. In fact, we have assessed a large number of diverse city, state, and business policies, covering not just energy generation but also transportation, buildings, the land sector, methane, HFCs, and others. A significant data and analytical effort went into assessing how these diverse and overlapping policies would impact overall U.S. emissions. Accordingly, in the text, we have sought to clarify in all instances what we are doing with respect to our analysis.

As one example, we have modified our abstract to provide a clear statement of what work we did: “We aggregated U.S. state, city, and business commitments within a global integrated assessment model (GCAM), to assess how a comprehensive national climate strategy can be built from expanded actions at subnational levels.” We also added the following statement at the end of our introduction: “In this paper, we describe how a significant subset of those actors in the United States—including states and cities, and some businesses—are implementing policies that significantly reduce economy-wide emissions, and how expanded action from those actors can provide a higher platform for potential additional Federal action.”

In doing this we believe there is one contribution of our analysis that we should clarify for the reviewer, and one area that we agree is a legitimate limitation of our analysis.

The first area that we believe was a matter of misunderstanding is that, while we assess all of these diverse actions from cities and states, **our methodology does have us aggregate these actions to the state level for purposes of undertaking an economy-wide assessment of emissions impacts.** In other words, **we run a 50-state model in which each state’s emissions are driven by a combination of both state-wide, and city policies.** As such, while our analysis ultimately is based on a 50-state assessment, the drivers of action are carefully assessed. **This is the function of the “Athena” model that we describe in the paper and the SI**—and which we have undertaken significant revisions in the text to clarify. **Athena is the structure within which we carefully eliminate the potential double counting that can arise from having, for example, a city policy that might be partially or fully subsumed by a similar state policy.** In light of the reviewer’s comments, we have clarified the language about our methodology, both in the main text and in the methodology section and SI.

In another way, the reviewer is absolutely correct. That is, we do not seek to incorporate actions of entities that are not states, cities, or businesses. For example, the “We Are Still In” coalition includes a number of specific categories of groups, such as tribal groups and universities, that have relatively few policy levers and a relatively small potential impact on emissions. We do not attempt to quantify the impacts of those types of actors. There are two reasons for this. First, there is not usually sufficient baseline and other data for us to make a confident assessment. **Second, it is also our view that compared to the city and state policies, the absolute impact of such activities would be small.** Business action is one category that could potentially impact the results, although likely in a smaller way, and we

can only partially cover that area. Unfortunately, most business reporting is done on a company-wide rather than a plant-by-plant basis, so the inability to pinpoint geographical location of reductions precludes a confident assessment of many business actions. **However we do evaluate a limited number of actions, which are listed in the summary table in the Methods and elaborated in more detail in the SI. These include 24 utility decarbonization goals, voluntary corporate actions through EPA Gas STAR, and voluntary HFC action through GreenChill. We evaluate these as so-called *pledged actions* and therefore do not include them in our Current Measures scenario. We do include these actions in the more ambitious scenarios, although they are frequently overridden by other, more ambitious assumptions.**

We thus view our assessment as a lower bound on the impact of subnational actions. At the same time we think that given other uncertainties, our numbers are robust. Because we are capturing what we believe are the most significant actions, we believe that the analysis does effectively illuminate the potential impacts of subnational action on the U.S. national emissions trajectory.

Response:

- In the text, the clarifications between NSA participation and those actors that are quantified now make this distinction clear.
- The authors should certainly include a discussion of the analysis's limitations, particularly as they result from omitting certain policies/actions, or actors, as described above.
- By 'most significant actions,' do the authors mean in terms of emissions impacts, or potential to contribute to national policy?

In this context, Fig. 1 is completely misleading as I don't see any evidence in the SI that tribes, healthcare organizations, faith-based organizations' commitments are actually analyzed.

Thank you for the comment. In this introductory position in the paper, we are focusing primarily on simply conveying the idea—which we think may not be obvious to people not deeply engaged in this field—that there is a broad diversity of action in the U.S. To this end, we believe that the map helps to illustrate a broader concept, which is the breadth of action across the United States, and thus we have retained this figure to help support the motivation for better understanding the diversity of this action. It was placed in the first section that highlights the overall trend of sub-national action and was meant to give the reader a sense of scale. At the same time, we fully agree with the reviewer that we don't want to mislead readers into thinking we analyzed every action by these coalitions. The dots on the figure do represent all actors, and as the reviewer points out correctly, some of these we did not explicitly model. As such we have clarified the purpose of the figure in the manuscript and figure caption. We have added the following clarifications:

Despite the scale of this groundswell of diverse new climate commitments, the central question remains how such coalitions, acting outside national government authorities, can impact overall national emissions trajectories—and, by extension, how broader action across countries can affect the global emissions trajectory. **The research presented here does this by estimating the impact of a subset of the diverse actors shown in Figure 1: the U.S. states and 285**

largest cities who have the most impact on overall U.S. emissions.

And in the caption to
Figure 1:

a) States, cities, businesses, universities, faith-based and other organizations now number greater than 3800. **This study assessed the most significant subset of these actors, focusing on states and the 285 largest cities.**

In evaluating the 50 states and 285 largest cities, we found 168 cities with policies that we were able to include. We describe this process in the text, the Methods section, and in the Supplementary Information. To be fully clear, we have also created a new figure, which is Figure 2 in the Supplementary Information, to show the geographic locations of the cities included.

Response:

- These edits make the selection process much clearer. Am I right in assuming the 50 states and 285 cities were assessed and (I assume) incorporated into the enhanced non-federal scenario, and the 34 states and 168 cities had quantifiable policies were included as part of the current measures scenario? This feels a bit nit-picky, but this could be one thing that's further clarified in the SI in the first section.

There's already a rich literature on state-level climate/energy policies that the authors fail to engage:

Rabe, B. G. (2008). States on steroids: the intergovernmental odyssey of American climate policy. *Review of Policy Research*, 25(2), 105-128. Peterson, T. D., & Rose, A. Z. (2006). Reducing conflicts between climate policy and energy policy in the US: The important role of the states. *Energy Policy*, 34(5), 619-631. Goulder, L. H., & Stavins, R. N. (2011). Challenges from state-federal interactions in US climate change policy. *American Economic Review*, 101(3), 253-57.

Thank you for these suggestions. We agree that the literature on state and non-state climate action goes back to important earlier works, and we absolutely want to converse with that earlier literature. In our previous draft, we did refer back briefly to some of this literature; for example, the Rabe 2008 paper was already cited and discussed in that version. Nevertheless, the reviewer's point is well taken. Accordingly, we have expanded the section referring to this literature more to more fully reflect how those ideas relate to our own work. As part of this, we have also integrated discussion of the additional two papers suggested by the reviewer.

We have also enhanced our discussion of the unique character of the particular circumstances in the United States today, with increased international engagement on climate change, severely diminished federal leadership, and an unprecedented level of explicit, bottom-up action on climate in the United States. This circumstance brings into sharper focus and raises the relevance more than ever before of

the role that bottom-up actors might play in climate mitigation. It is this emerging circumstance that provides the motivation for this paper.

Response:

- The context of the circumstances around US climate action is a helpful addition to the framing of the paper, although see my comment above about the paper's overall framing.

The authors primarily frame U.S.-state level action as a net-positive, which I think is largely correct. But this is an assumption that merits some further discussion. For instance, the US has also seen some push-back against climate action at the state level (e.g., Ohio's HB 6 law, which subsidizes coal and guts clean energy programs) -- what is the risk of other forms of this kind of push-back based on shifting political leadership? Could state-level action also be weaponized against climate action, if, e.g., there is a 2020 administration committed to ambitious climate action?

This is an incisive point that raises some important political questions, and one that we agree with fully. We have added some text to reflect this issue in the paper, specifically:

It is important to note that while the aggregate effects of sub-national action may serve to build a higher overall level of action in the United States, individual sub-national actions are not always consistent with long-term climate goals or clean energy transitions; they may also lead to conflicts and economic inefficiencies. States have sometimes also opposed federal climate policies through court actions.

Response:

- These additions in the most recent draft address this comment.

Methodological
comments:

3

The authors seem to contradict themselves in the section 'Assessing the current and potential impacts of diversified climate action.' Again, see my comment about the conceptual framing above - the authors should just be straightforward and clear - they are using a state-level policy model to assess national-level scenarios. They seem to dance around some of the challenges of fully incorporating 'bottom-up' action in this section in way that suggests that their model does something (i.e., assess "stakeholder engagement, data collection, aggregation, and modeling" p. 5. Lines 21-23) that other IAMs and GCMs are unable to do, but I don't see any evidence of that.

7

Thank you for the comment. We addressed the bigger picture framing question above, as well as how our approach does provide a deeper assessment of bottom up action than previous work. There is a second dimension to this comment here that we can respond to here. Our model is really in three parts, and there is a fourth element that the reviewer mentions here. First, we do have a data collection component of the work, which is both challenging but is an essential part of the contribution of this paper. We obtained data on subnational goals, policies, baselines, and targets from a number of sources, including CDP, Under2MOU, C2ES, ICLEI, carbonn, ACEEE, Second Nature, BEA, U.S. EPA, U.S. DOE (including the SLED tool), and others. Data sources are cited in our sector-level descriptions in the Supporting Information. **Second, we then place these inputs in a context of a broader state, national, and global baseline, and then carefully aggregate the impacts on that baseline up to the state level. That is the function of our Athena tool, which is calibrated to an overall GCAM baseline.** Third, we then take these **state level inputs (i.e. the impacts of subnational policies at the state level)** and run them in GCAM to account for economy-wide interactions across the national and global climate, energy, and land systems. **That three part process is novel for IAMs.**

The other dimension mentioned by the reviewer is stakeholder engagement. This did not impact the modeling strategy above, but instead was used to establish our scenarios. Rather than just make assumptions for subnational action, our process engaged with actors to see what they found to be the most plausible policies to expand at the subnational level. We therefore think that adds an important dimension for the integrated roadmap strategy that is exemplified by this research. However, we do not have space to really describe this process in detail for this paper, so we just mention it briefly. It might be valuable to do as a separate white paper or publication.

In response to these helpful comments, we have added additional material on our overall modeling process and the Athena model in particular, as follows:

P.4 - Clarification of methods To address these challenges, we developed a new, comprehensive strategy to assess the scope and **emissions impact of non-federal climate actions in the United States. We do this by integrating data on baselines and sector-specific policies and commitments** into an economy-wide estimate of the U.S. emissions trajectory to 2030. The overall process involved steps of data collection, baseline development, aggregation, and modeling, with parallel stakeholder engagement to help develop the policy platform for scenarios (Described further in the Methodology section and in the SI). The resulting approach combines detailed accounting of bottom-up actions with an integrated assessment model to understand their overall emission impacts.

Response:

- In the SI, Tables 3 and 4 are helpful in understanding exactly how and where state, city, and companies are factored into different scenarios.
- The reframing of the overview in the SI is also helpful in clarifying how the analysis process occurred - particularly the definition of Athena as “a new set of bottom-up models (collectively

called Athena) to aggregate non-federal climate policies into state or regional activity data for each sector.” It might be helpful to apply that same framing - of Athena as a set of models rather than a single tool in the main body of the text - e.g., on page 6 and at the top of page 16. See below other comments on Athena.

There is mention of an “Athena” tool but it is not at all described in very much detail in the SI (mentioned briefly in a few paragraphs on p. 2), but it’s not clear what “effects of different actions” (p5., line 29) are included. Why not provide a table in the SI with these details to make it 100% transparent and clear what is actually being evaluated? I found this entire paragraph on page 5 about Athena and it’s integration into GCAM extremely vague and unclear.

Thank you for the suggestion. We have added additional material in the SI to clarify what Athena is and how it works, as well as how it interacts with GCAM. In addition, we agree that the goals and purpose of Athena needed to be made more explicit in the main text. Accordingly, we added the following on p. 5:

In assessing actions that impact emissions with overlapping or nested geographical coverage—for example, a city policy on renewables within a state that has a renewables target— there is a need to eliminate double-counting and otherwise account for potential interactions in the areas of overlap. At the same time, it is also important to ensure that baselines and input data across jurisdictions are consistent to ensure that the resulting broader projection is robust. To address these issues, we developed an aggregation methodology (which we call Athena) that draws from a well-established set of inputs from GCAM and then aggregates the effects of diverse climate policies while explicitly considering interactions and avoids double counting within each sector.

Here is some specific language that points to what I mean:

Response:

- This is a helpful clarification, but again, it would be helpful to ensure Athena is described consistently across the text. It doesn’t appear that Athena is a model or tool as it was initially described (or at least how I was picturing it, similar to an IAM or scenario tool). The authors’ clarification is helpful, especially because the Athena methodology is not published elsewhere (as is GCAM on the GitHub link provided) nor is it peer-reviewed. From what I can tell from the added description, figure, and table, it appears to be a way of determining which policies to

include due to potential overlaps, which then allows for some modification of GCAM baselines. If this is the case - then why present this method as a model or tool or try to frame it as such, and why give it a name? I don't think that giving a methodological process a mythological name like Athena is actually useful - if anything, it obfuscates the purpose of the paper and makes the reader question the validity and credibility of the GCAM scenarios and outputs. The authors should just explicitly say that they instituted a decision-making protocol for aggregation (similar to Kuramochi et al., 2020) and include those decision-points in a diagram (which in my mind should be a flow chart instead of what is shown in Figure 5's oversimplification of what they had written in the previous text "Athena Delivers appropriate aggregated inputs back to GCAM-AP").

- It seems clearest to me as it's initially described at the start of the SI: as "a new set of bottom-up models (collectively called Athena) to aggregate non-federal climate policies into state or regional activity data for each sector" that's part of the broader 4-step methodology described in the response above.

"Explicitly considering interactions to avoid double counting" - ?? what do the authors mean by 'interactions'? It seems they're framing them as negative but they could also be positive (e.g., amplifying, see Chan et al., 2015)?

We address potential issues with double counting in two ways. First, we account for overlap between policies within a sector in Athena (for example, cities within a state that both have a RPS). Second, we model interactions between sectors in GCAM (for example, impacts of electric vehicle policies on electricity demand). We have added further discussion of our process for accounting for overlap in the SI (see *Avoiding Double Counting*). To avoid confusion, we changed "interactions" to "overlap" in the sentence referenced by the reviewer.

We recognize that in practice policies can be overlapping or reinforcing in terms of impact. We discuss how we account for overlap on a sector-by-sector basis in more detail in the *Sector Descriptions* section of the SI. For example, for RPS policies:

We aggregated renewable energy generation resulting from the state, city, and utility actions described above at the state level, partially accounting for overlap across the three different levels of action. We accounted for city policies using a net percentage approach, where additional renewable demand from city goals in percentage terms (beyond the percentage required by state policies) is added to the state total. For example, a city with a 50% goal in a state with a 40% RPS add a net 10% renewable generation to its electricity load. Alternatively, if the city's projected renewable target were less than the state's 40% target, no additional renewable demand would be included. We applied the same net percentage approach when accounting for utility targets.

Response:

- These changes make the discussion of overlaps and double-counting much clearer. Ideally, the authors could actually include these descriptions and decision-points in a flow-diagram revisioning of Figure 5.

“Delivers appropriate aggregated inputs back to GCAM-AP” - this statement is also too vague and not reproducible.

We agree this statement is vague, and was also unnecessary. We have deleted it.

“Accounting for overlap” - how? What overlaps? Where? Between actors/sectors/emissions?

Here we meant between actors -- for example, a city within a state that both have a renewable portfolio standard. We have clarified this in the text.

Again, ideally the authors could actually include these descriptions and decision-points in a flow-diagram revisioning of Figure 5.

5

“Athena generates estimates of impacts in terms of activity data appropriate to that sector” - is this done on an actor basis, state-level?

This is done on a sector level. We clarified this.

A diagram that explicitly makes clear the flows between Athena and GCAM would be 100% helpful here. The authors should remove Fig.1 for being misleading and instead actually provide a diagram of the interaction between Athena and GCAM-AP. Also, Athena has not been published or peer-reviewed, from what I could tell from a web of science search, so I question it's rigor or what it's doing, especially because the authors provide scant details on what it includes, how it's doing its “interaction consideration” and “avoiding double counting” and “appropriately aggregating inputs.”

Thank you for this suggestion. We have created a new diagram illustrating the modeling process and feedbacks (in the Methods section of the main paper).

Would like to see a clear listing of exactly what policies are included and their estimated emissions reduction impacts. It's unclear from the SI what specific policies are being included and analyzed. In several places, the authors mention "and many other policies," - this is vague and not reproducible.

We have now added a table listing the policies included in each scenario (in the Methods section of the main paper). We describe these policies in more detail in the *Sector Descriptions* section of the SI. Since we model emissions reductions across the U.S., we do not estimate emissions reductions resulting from individual policies. However, we have included a new figure in the main text that breaks down emissions reductions contributions by sector.

Response: The additional tables are extremely helpful.

The authors' approach to measuring Tier 1, Tier 2, and Tier 3 states, based on their likelihood of adopting climate action is a potentially very helpful contribution to attempts to predict the impact of climate action in the US. More details on the criteria that informed their classifications (e.g., how exactly, the outlined criteria for placing a state in each tier were defined) could be useful to other analysts focused on non-state climate action.

Thank you for this suggestions, we have added a listing of the criteria in the main text:

In parallel, we generated three groupings of states that reflect the likelihood of accelerated action, based on attributes such as membership in climate organizations, vocal leadership in support of climate action, ambitious emissions reduction targets or standards, and on-the-books climate policies. In our scenario, Tier 1 states realize levels of ambition roughly equivalent to today's leading states like California, Tier 2 states lag but still move toward the level of ambition of Tier 1, and Tier 3 states which do not implement major changes (SI).

Response: This description is helpful; a rubric outlining some of the thresholds and sources for evaluating these criteria would be nice/useful (perhaps something that could be produced as part of future outputs?), but I'm not sure it's necessary.

The baseline remains unclear - the potential reductions of the current measures, enhanced non-federal measures and comprehensive measures are compared to the NDC target. At times in the SI it seems as if the current measures scenario is used as a kind of baseline, but the implementation of existing subnational commitments is far from certain, and, I would argue, worth understanding on its own terms and for its own contributions from a BAU pathway. The assumptions underlying the trajectory of national emissions in this scenario could be made clearer and more detailed. Page 7 reads: "Figure 2 shows the

results of each of the three scenarios. Current Measures would deliver reductions of 25% below 2005 levels by 2030. In contrast, absent any climate policy action, we estimate that absent policies, economic growth would lead to a roughly 4% increase in national emissions in the United States.” More details about this baseline scenario (does it project current emissions trends forward based on economic growth? Does it assume current federal policies continue to be implemented?) - in both the text and in Figure 2 would help the reader better understand the potential impact of bottom-up climate action.

We have added a new section to the SI to describe our reference scenario in further detail:

Initial data from GCAM-AP was generally interpreted in Athena as a no-policy, reference scenario in which subnational policies, and some key federal policies, are not represented. Thus, the full impact of policies was applied to the baseline projections without needing to address overlap. Exceptions to this assumption and cases where any subnational policies were already embedded in the baseline are discussed in the sector descriptions in Section 5.

The GCAM-AP reference case scenario does include certain federal-level policies that have significant impacts within the sectors modeled. These include the federal production tax credit (PTC) and investment tax credit (ITC) in the renewable energy sector and federal fuel economy standards in the transportation sector. While sectoral modeling results in Athena typically represent the impact of state, city, or business policies only, final modeling results from GCAM-AP account for the combined impacts of these federal-level policies and the non-federal impacts from Athena. In addition, two types of federal policies not already included in GCAM-AP were explicitly modeled in Athena and aggregated with non-federal actions before being passed back to GCAM-AP as inputs. These were the U.S. EPA Section 608 refrigerant management policy for HFCs and regulations to reduce fugitive emissions in the oil and gas sector.

Response: This is a helpful clarification. One question about the added text: in the sentence “*While sectoral modeling results in Athena typically represent the impact of state, city, or business policies only, final modeling results from GCAM-AP account for the combined impacts of these federal-level policies and the non-federal impacts from Athena*” does the text “these federal-level policies” refer to the “*federal production tax credit (PTC) and investment tax credit (ITC)*”?

How do their assessments compare with other assessments of U.S. “enhanced ambition” or re-engagement of the federal government? Would like to see in the discussion some engagement with other analyses and literature (i.e., the Rhodium Group, Kuramochi et al., 2017 and Kuramochi et al., 2019). No references are made to these other pieces in either the main text or the sensitivity analysis that also analyze future potential emissions reductions from U.S. subnational/non-state actors, which are very similar in their analyses.

There have been a few other assessments of the impact of subnational action on emissions both at the global level and, in a couple of cases, in the United States specifically. We discussed the global

perspectives in our framing sections, but the reviewer makes a good point about two analyses that look specifically at U.S. emissions under current measures. Kuramochi and others are co-authors on the two studies that we noted in the first draft, and we have now added explicit reference paper suggested by the reviewer as well as one new one. Substantively, this group is strong and we agree they have done good work looking at global impacts in particular. Because their focus is global, their assessment of the U.S. is necessarily less detailed. Their numbers are roughly in line with ours, but their aggregate approach does not allow for the explicit linking of policy roadmaps that our more detailed analysis allows. In this sense, it remains our view that the methodological approach we present here is an important advancement. Also, our are broadly consistent with Rhodium Group’s study from 2017, although their lack of published methodologies makes a detailed comparison of comparability between their results on ours somewhat challenging. We have added a reference to this in our paper, because we agree with the reviewer that some comment is helpful here. We have added the following to the text:

There is currently no published estimate directly comparable to our higher-ambition *Enhanced Non-Federal* and *Comprehensive* scenarios. On the other hand, the estimates for the impact of existing policies, including state and city actions (i.e., the *Current Measures* scenario), are broadly consistent with coarser estimates that have been made for the potential of U.S. emissions reductions. Kuramochi et al., for example, assessed the impact of current climate policies on 25 key emitters, including the United States. Their global assessment was necessarily less detailed in its roadmap for how climate action could be implemented in the U.S. from sub-national action, and the best analogue to our assessment is to our Current Measures projection. Their range of outcomes in 2030 shows a trajectory for the U.S. to achieve roughly a 15-26% reduction, consistent at the upper end with our central estimate of 25%. Similarly, in 2017, the Rhodium Group used publicly available versions of GCAM and NEMS to forecast near-term U.S. emissions, and they projected a 15-18% reduction in 2025 and a 14-19% reduction in 2030, relative to 2005. Our Current Measures results are slightly higher than these, which we attribute to the fact that those estimates do not fully include the sub-national actions that our estimate includes, and they also pre-date much of the recent wave of bottom-up climate commitments described earlier

Response:

- The added text addresses the original comment.

Other notes:

- Small typo in “Paris Agreemnt” (Page 4, line 16)
- On page 7: “In contrast, absent any climate policy action, we estimate that economic growth would lead to a roughly 4% increase in national emissions in the United States.” Am I right in assuming this means absent any additional climate policy action?

Reviewer #2 (Remarks to the Author):

This paper discusses a trending issue to provide the possible mitigation sub-national actors might deliver. The modeling based on GCAM USA and Athena provides reliable quantification of how these sub-national actors can contribute to the climate actions.

Thank you for the comment and reflection on the relevance and reliability of our analysis.

However, the detail has not been clarified enough for the reader to understand how many firms or cities have been involved in this study, what kind of action has the best effect, etc.

We have provided more detail about this now in the paper, and have enhanced our descriptions of this wherever possible.

The authors raised two interesting questions in the beginning, on 'What bottom-up actor groups could deliver? And how these groups could support greater national actions?' I think more results could be included to discuss these two questions and your findings.

Thank you for this comment. We have added some additional results into the main text, including a new figure, to provide additional context for and support of our argument.

My detailed comments are listed below

Page 4 line 9: How do you eliminating overlap for the population and GDP? The number seems very impressive that 65% of the US population has been engaged in climate actions, but I'm not very certain if you can detect if one person been engaged in several sub-national coalitions? (I've also roughly go through the report you cited, perhaps I have missed it, but didn't see how they eliminate overlap)

Here we are calculating the percent of the U.S. population whose subnational governments have declared support for the Paris Agreement. We avoid double counting by not counting the population of any cities within a state that has also declared support. We have added further clarification in the text.

Page 5 line 29: I would suggest you include more detail on the Athena model if this is one of your innovations. The underlying theoretical mechanism needs to be explained here. And I also recommend you to include 'Figure 2 ATHENA Modeling Flow' from your technical appendix of 'Fulfilling America's Pledge' in the supplementary materials.

We have added a paragraph in the main text describing Athena, and have improved and included the figure suggested (Figure 2 Athena modeling flow). It is now in the Methodology section of the main text.

Page 6: The name of the scenario could be better unified long the text (Bottom-up/All-in vs Enhanced non- federal/Comprehensive).

Thank you, we have updated all of our scenario names to be consistent.

Page 6 Scenarios: It would be nice to clarify the scenario difference more, either in the explanation of scenario setting or in the supplementary material.

We have added substantial elaboration on the scenarios in the main text Methods and in the SI. We describe the conceptual differences in the main text and in bullets in the Methods:

- *Current Measures projects where the U.S. is headed given current policies at national, state, and local levels (Table 1).*
- *Enhanced Non-Federal examines outcomes under a broad expansion of cutting-edge climate policies on the part of states, cities, and businesses (Table 2).*
- *Comprehensive explores a climate strategy integrating aggressive non-federal climate action with renewed federal engagement after 2020 (Table 2).*

Table 1 and 2 provide descriptions of the specific actions included in our analysis. We also included more descriptions of our methods in the new *Scenario Development* section of the SI as well as in the *Scenario Descriptions*.

Page 6 line 18: For enhanced non-federal scenario, I'm a bit confused about why you classify the level of potential state ambition and how it works for Athena.

We group states in different tiers to reflect their willingness to engage in climate action. We identified these states by attributes such as membership in climate organizations, vocal leadership in support of climate action, ambitious emissions reduction targets or standards, and on-the-books climate policies. We added more detail on this in the main text when we describe the *Enhanced Non-Federal* scenario.

Page 7 Figure 2: How do you address the uncertainty for the three scenarios? Why the emission trajectory (Figure 2) have a range of uncertainty but the sectoral results (Figure 3) don't have?

We added further discussion of the treatment of sensitivities in the main text:

To better characterize the range of possible emissions reductions of the scenarios in this study, we examined each scenario under varying assumptions about socioeconomic change, technological change, fossil fuel prices, and the size of the land sink. Based on these sensitivities, reductions in the Comprehensive scenario could be as low as 46% or as high as 52%. While these are only a subset of uncertainties, they provide a range of possible emissions associated with different levels of climate action. (See Methods for further discussion.)

9

We also add discussion in the Methods:

We also examine economy-wide results across a range of values for important drivers in our analysis. These include economic growth, population growth, oil and gas prices, renewable energy costs, coal power plant retirements, and the magnitude of the land sink. While these represent a subset of all possible uncertainties, they give a partial indication of the range of possible emissions for each scenario in our analysis. See Table 2 in the supporting information for a list of assumptions and sensitivities for the economy-wide analysis.

We further discuss these in the SI. For visual simplicity, we include our sensitivities in our line graph showing emissions reductions but not in the sectoral results.

Page 8 Figure 3: Could be better using transparency to differentiate the historical data and model results for Figure 3 (a). We added clarification in the caption that the grey line indicates historical data.

Page 9 line 30: 'The next year' seems to be quite rough. It would be nice to notice which year it is (as I suppose is this year, 2020, now?).

We have changed the text to "This is a critical period" - given that now the timeline for submitting new NDCs may spill into next year, but no decisions have been made yet, we thought it best not to anchor to a specific year.

Page 9 Discussion: You made a very strong argument on why we need to focus on the sub-national actions and I totally agree these were more important than ever for the time being. Yet I would suggest you include more findings from your model to support this idea, with such a big model like GCAM, it would be such a loss to show only such few results.

Thank you for the comment. The reviewer is correct that GCAM produces a vast quantity of outputs, many of which are interesting and worth discussing. So we are happy to have the invitation to share more results. Mindful that we still have a space constraint, we have added one additional figure and some associated text: “we have also prepared the sectoral policy numbers by state that serves as a building block to the state-level modeling. These files are attached for the review, and will be made available upon request (due to the inclusion of proprietary data).”

This figure shows a sectoral breakdown of where the emissions reductions come from within our model framework. Combined with the deeper dive on the power sector, we think this gives a more complete picture of our results.

1
0

Response to Referees NCOMMS-20-08589B

"Fusing subnational with national climate action is central to rapid decarbonization: The case of the United States," Hultman et al.

Author Response to Reviewers

Overall: We are very grateful to Reviewer 1 for the additional perspective on the framing and for the suggestions for improving the paper. We have undertaken a revision of the paper that we hope addresses the reviewer's comments effectively. This includes:

- A re-focusing of the contribution of the paper toward an assessment of the *impact* of subnational action rather than also referring to the mechanisms by which such action can occur.
- A clarification that the novel dimension of our analysis is a quantitative analysis of state, city, and business actions, and its implications for how we understand the construction and delivery of national climate action strategies. This has not been done before with this level of rigor.

A logistical note. We kept the earlier conversations with Reviewer 1 and Reviewer 2 in this note for context, but for clarity have put a gray highlight over all of the old discussion. The new comments from *Reviewer 1 (green)*, as well as our responses (black), are in white background.

Reviewer #1 (Remarks to the Author)

Overall, the piece reads more like a commentary/perspective rather than a research article. I was already familiar with the white paper, "Accelerating America's Pledge: Going All In ...," but I'm not sure that paper, which seems an adaptation of the report, warrants a research article in *Nature Communications*. I do think if the authors shortened the piece and focused on providing some commentary/discussion on the likelihood or factors that might lead to certain scenarios they discuss, it could be an interesting commentary. But there are unfortunately both conceptual and methodological issues that are serious enough to give me pause for its recommendation as a research article.

We thank the reviewer for the many helpful comments and have undertaken to revise the paper in a way that we hope frames our analysis more clearly and also answers the points about the broader contribution. With respect to this specific comment on the tone of the paper, we agreed and have undertaken some restructuring to highlight the research contributions more upfront, and have moved the more broad elements to the discussion section.

Response:

• *While the authors have provided more context for city and state-level climate policy in the U.S., there is still a major gap in the conceptual framing of the paper that remains unresolved. The title of the paper remains "diversified climate action," which the authors describe in the introduction to mean diverse coalitions and groups of subnational and non-state actors. While this topic is of timely interest given the lack of national U.S. leadership on climate change and energy, the paper still remains primarily focused on state-level U.S. climate action, mandatory or binding policies (in the authors' own words), and broad, sector-level actions that are modeled at a highly-aggregated, sectoral level for a picture of U.S. emissions under three scenarios of varying ambition levels. The primary question I kept asking myself in re-reading this paper and the SI is the question of how much of the policies evaluated (i.e., renewable portfolio standards, energy efficiency standards, coal phase-outs) really reflect a "new diversity" of "bottom-up actor groups" as opposed to what states have been doing all along.*

We agree fully about making sure that we are not insinuating or claiming that this diverse, non-national action is new, nor that we are capturing the governance mechanisms with this particular analysis. We further agree with the Reviewer that our previous framing conflated two important aspects of the phenomenon we are studying. First, "does climate action happen from the bottom up." Second, "is the expanded bottom up action of the past, say, five years providing a basis for potential reductions that are, by some definition, 'significant'?" These two pieces should be

separate, and your comments last time and this round have helped us clarify this for us. To remedy this, we have removed references to the “newness” of the phenomenon and now focus primarily on how the empirically expanded and deepened set of actions, including those from cities and some businesses, could meaningfully impact U.S. emissions. We believe that this will address the Reviewer’s suggestion.

We would also like to emphasize that the approach in this study is built around a bottom-up analytical approach that includes cities, states, and businesses. For the current measures, we have estimated the implications of the commitments and actions taken across all 50 U.S. states, the 285 largest cities (those with populations over 100,000 or have readily-available data on climate commitments), and a diverse set of corporate actors. Our two-step analysis process allows us to analyze these at a sectoral level first, using a bottom-up accounting model, and then to look at the interactions across sectors second, using an integrated assessment model with 50-state detail. At the same time, when we look at the two enhanced ambition scenarios, we have expanded action at a sectoral level using states as the organizing framework. The tiering approach we use, in which we group states into three tiers with indicative levels of ambition and associated policies, uses state boundaries as a proxy for estimating overall ambition within that state. Whether this implies only state-level policies or a combination of state, city, and business policies is not defined in our analysis, but it was built to map to the level of combined city, state, and business action that we are modeling in the current measures scenario.

In the U.S. context, climate policy has always been driven primarily by states - California is a prime example and the policies included in the ‘Current Measures’ scenario are reflective of states’ leadership on climate policy. In fact, the U.S. has never had a comprehensive, nationwide climate policy. The authors identify key gaps with respect to bottom-up climate actions on p. 2 lines 32-34 that there is limited understanding of what bottom-up actor groups could deliver and how they could support greater national actions. They also claim on p. 3 lines 5-7 that the paper “describe[s] how a significant subset of those actors, including states and cities, and some businesses are implementing policies that significantly reduce economy-wide emissions.” I unfortunately still feel that while the paper has much potential to accomplish these goals, the article as it currently stands fails to achieve these goals.

We’re glad for the vote of confidence in the underlying materials and hope that our revisions, described below, will enable us to match our contribution with the correct framing.

The authors do not answer the ‘how’ part (i.e., it would be fascinating if the authors could disentangle which policies, particularly if implemented at the business or local government level, could actually build the meaningful catalytic chain of policies that would “support greater national actions”, similar to what Martin and Saikawa (see the reference below) do in their piece examining state-level policies’ effectiveness at driving down power sector emissions).

We agree the Reviewer makes a good point here and we have retooled our narrative to be in line with this comment. We discuss this further below as well. We agree that our contribution in this paper is more about the impact, and less about the mechanisms of identifying the catalytic chain, although we ourselves find this a fascinating question. In any case, this topic is a large one and we view this angle as beyond the scope of this particular analysis. Nevertheless, because of the importance of this question, we continue to refer to other work in this space, and added in a paragraph in the discussion about gaps and future work that includes this.

In terms of the ‘what’ part of bottom-up groups contributions, the greater detail provided in the SI still appears to be primarily focused on state-level policies. It would be helpful to understand the magnitudes of these contributions from various policies and actor groups so that a policymaker could understand the relative impacts of different groups and policies. I’m reminded of Meckling et al. (2015)’s piece on winning coalitions for climate and energy policy - a detailed analysis of how different actors engage to catalyze policies at the local levels and within the private sector to generate support for national-level policy could be a more compelling framing. But this article still primarily focuses on highly-aggregated, sector level contributions and doesn’t go into this level of granular detail.

I'm not saying that this article doesn't have value. It does, but it's still not framed appropriately and as such, it falls short of delivering on its current promise. If the authors would simply reframe the article as an investigation of state-level climate and energy policies' impacts and contributions in a current measures scenario, and then what these impacts could be if more states (or if there existed a national policy that applied to and was implemented by all states) adopted and implemented them, then this article to me would be a lot more conceptually coherent in the context of the very valid research question of how the U.S. can continue making progress on climate change in the absence of coordinated U.S. action.

Thank you for the suggestion. We have retooled the paper's framing according to this suggestion. As we noted above, we agree fully on the importance of distinguishing a few key ideas in the framing, making sure to focus our contribution on the elements that Reviewer 1 notes in the next section. Accordingly, we have revised every part of our discussion across the entire paper that relates to the framing. We have done this in a number of ways, some of which we reproduce here to demonstrate the types of changes that were made.

- First, in the abstract, we revised to better signal that the idea of bottom-up action is not new, and that our analysis is focusing on the expanded action in the U.S. and whether it can be a basis for genuinely deep emissions cuts. We have also narrowed the framing of our contribution to be more on this topic, and then suggest its relevance for broader discussions of the kind that the Reviewer notes.
- To address the reviewer's concern with the title that includes "diversified climate action", we have changed the title to: "**Fusing sub-national with national climate action** is central to rapid decarbonization: The case of the United States"
- In the introduction, we have made a number of changes, again to show that the idea of bottom-up action is not new, nor is it our contribution in this paper. Instead, we present it as an idea that has not been tested at this expanded scale nor studied in the way we present in this paper. For example:
 - "and such actions are now becoming sufficiently broad and ambitious to challenge **conventional** understanding of both the models for achieving global success and the scale of their potential impact. While climate policy has often been perceived as driven by national governments—**particularly by governments themselves**—the potential value of this "bottom-up" action in delivering higher global ambition to reach 1.5°C compatible pathways may be much greater than has been realized.
 - **This principle of diversified, bottom-up action has been noted by advocates, the research community, and subnational leaders as an important basis for climate policy for decades** and formed the basis for organizing coalitions of climate actors even well before Paris Agreement.
 - Deleted reference to "how these groups could support greater national actions."
- At appropriate points in the paper, we have also changed the word "Diversified" to "Subnational" which we think is a slightly better and narrower focus. We have grappled deeply with the question of terminology over many years and have yet to find a perfect term, having weighed terms such as diversified, subnational, non-Party, non-State, non-Federal, and the technically correct but awkward "state, city and business actions". We tried using "diversified" with this paper for simplicity, but agree with the Reviewer that our analysis is not about the full set of "diversified" actors but only an important subset. "Subnational" has its problems as well, as some businesses are multi-national and many governors and mayors resist being called "sub" or "non", but for the paper it's a pretty good fit to the states/cities angle, and also connotes a governmental aspect. As a guidepost we have generally retained the term "diversified" for the Aristotelian ideal case of "a diverse set drawn from non-federal and federal actors" and applied the term "subnational" to the specifics of our analysis.
- In the section introducing the U.S. case:
 - Renamed section heading to *A case of diversified climate action: **State, City, and Business Actions in The United States***
 - Text revision: "**The long-term evolution of state policies in particular has consistently pointed toward their potential for driving national emissions**"

reductions even while national climate policies were absent, scattered, or delivered without national legislative approaches”

- In the section discussing our case
 - Retitled section heading: Assessing the current and potential impact of **subnational** climate action
- In the discussion section
 - Restructured section to focus on first, our results, and then, how our results resonate with, though do not demonstrate, governance processes. Significantly rewrote the pieces to be clearer about our contributions. Condensed.
 - Test revision: “While it has long been expected that successful climate action will require commitments from national governments and appropriate international coordination³¹, the insufficiency of current response to climate change clearly points to a need to draw **more from broader models** of organization, coalition building, and implementation.”
 - Text revision: “As countries and their citizens evaluate options for more ambitious strategies for sustainable development and climate-friendly economic opportunities, decarbonization at the national level will necessarily be rooted in actions by a diverse range of actors, both governmental and non-governmental, including cities, states and provinces, businesses, and a wide range of other actors. **Our analysis shows that such actions have the potential to aggregate to meaningful reductions in line with global climate goals.**”
 - Text revision: “**grounded in other work on governance and policy, that suggests that such action may have essential benefits to the long-term robustness of action, as well as to the effectiveness of implementation. Specifically, the process of generating more diversified policies and commitments in itself could stimulate broader engagement for decisions that are more directly tied to constituents**”
 - Text addition: “This work has some limitations, some of which point to areas for future work. This work focuses on what we viewed as the key subset of actions from states, cities, and businesses. Largely because of data limitations, we did not include many non-state actions, which means we were unable to assess the magnitude of impact from those actions. This data limitation is significant for many actors in the U.S. and is even more acute for large parts of the world, which points to a need for additional data but, absent any organized national or global data collection strategies, the constraints on data collection and reporting are acute for many actors with limited staff capacity and budgets. In addition, our analysis is focused on emissions impacts, but a related and key question that our work raises for additional research is to better elucidate the social and governance mechanisms by which bottom-up action can generate deeper and more robust long-term results. This is an area of active research and given the potential centrality to achieving global climate goals will require significant new funding and support. Finally, our analysis required advanced modeling tools and a large re-search effort; it would be difficult to replicate for many national or sub-national level processes. A strategy to identify opportunities at national levels to tailor and simplify the strategy to national circumstances would be essential for making such approaches more broadly accessible.”

A supplemental comment. We feel a key part of our analysis is the fact that we did not simply look at state policies, but included cities and businesses where feasible. In addition, a different key aspect to our analysis is that the hypothetical national policy is actually based on the sub-national policies, as opposed to most studies which invoke some version of an economy-wide carbon price.

Even more meaningfully, this type of analysis could then allow for policymakers to envision the types of reductions possible with nationwide, coordinated U.S. action. In many contexts outside of the U.S., for example, we can see how national-level policies have been effective in achieving climate goals (i.e., China, Costa Rica, among others). The “bottom-up” part of state-driven climate policy in the U.S. represents a failure of national policy in some sense.

Yes, we fully agree and thank you for this perspective. Based on the good work that many have done to look at how large changes happen in the U.S. (in climate, and other policy areas too), it is clear that the “laboratories of democracy” model of state, and even city etc. actions, can provide a more robust groundwork for an effective and lasting national climate policy. Yet it has also been the case that in the U.S., many advocates as well as researchers have frequently mooted the “moonshot” of a single, federally imposed, economically efficient solution (e.g., tax, pricing) and foundered on the rocks of reality. We believe that our results can effectively demonstrate that a bottom-up strategy matters to overall emissions, and that is one (though not the only) piece of evidence to support better integration of this approach with an eventual federal strategy. We have added a reflection on this concept in the discussion.

Here are a few specific lines that illustrate what I mean by problematic framing of the paper:

P. 5 Lines 15-18: “Despite the scale of this groundswell of diverse new climate commitments, the central questions remains how such coalitions, acting outside national government authorities, can impact overall national emissions trajectories—and, by extension, how broader action across countries can affect the global emissions trajectory.” Characterizing predominantly state-level climate policy doesn’t answer the question about how “diverse new climate commitments” and how “such coalitions, acting outside national government authorities” can impact overall trajectories. The US’s climate trajectory has always been impacted by states, so I’m not sure what else is being added that is new or helps shed light on this question.

Thank you for the comment. We have removed the sentence.

P. 3 Line 18-19: “The United States today provides a globally critical testing ground for essential questions on how subnational commitments can contribute to better outcomes for overall national climate strategies.” - ok but major caveat that the U.S. context is really specific to the federated U.S. system - there are limits to the applicability. The authors should discuss this limitation.

We agree. We added a sentence in the next paragraph: “Also important is the fact that the U.S. political system is federal, which devolves significant policy authorities to state and other subnational government levels (cities, counties, etc.); we might expect more potential impact from actions in federal systems compared to systems that do not devolve authorities in the same way.”

P. 3 Lines 28-29: “The evolution of state policies next to Federal policies in the United States, has raised the prospect of national climate policies driven by enhanced “bottom up” action.” Again, U.S. climate policy has always been driven by states.

Thank you for the comment. We have clarified the sentence to read, “The long-term evolution of state policies in particular has consistently pointed toward their potential for driving national emissions reductions even while national climate policies were absent, scattered, or delivered without national legislative approaches.”

P. 6 Lines 33-36: “ In order to keep modeling conservative and analytically rigorous, the scenario did not include commitments to reduce GHG emissions on their own unless backed by a binding mechanism (e.g. the Regional Greenhouse Gas Initiative (RGGI) and California’s cap). A summary of these policies is provided in Table 1 and further details are in the SI.” These are again, mandatory, legally-backed policies, which are distinct from the largely voluntary nature of private sector, bottom-up action. This would not be problematic if the paper’s framing were recast as an examination of state-level climate policy and future contributions.

Thank you for the comment. In line with the spirit of this comment and others, we have reframed the paper about the impact of state, city, and business policies. We would offer one additional clarification on this particular sentence. The distinction we are making in this sentence is not

about state vs. non-“state” (US state) action. It is instead about the kind of action taken by states, cities, or businesses. For example, at a state level, a governor can profess a goal for net zero by 2050. More robust is a legislative mandate. We chose to model only the latter in our Current Measures scenario. Similarly, at a city or municipal level, a mayor can set a goal, or a council can set a legislated policy. We chose to model only the latter. Reviewer 1’s comment suggests that we are only modeling state policies. We are modeling state policies, city policies, and some business policies in current measures; the other two high ambition scenarios have more comprehensive coverage that includes largely voluntary actions by states, cities, and businesses. We have removed the parenthetical examples of the two state policies to avoid confusion; the reader is invited to review the SI for the broad suite of policies and how we judged the “bindingness” of each.

P.12 Lines 10-13: “The work presented here elucidates in more detail a mechanism for such interactions, providing opportunities for diverse actors to generate their own goals and commitments as part of a national process and to better understand their contributions to the overall emissions out-comes.” The mechanisms are not clear - again, the policies and actions are aggregated to the state-level so it’s not clear what mechanisms the authors are referring to.

Thank you for the comment. Throughout the paper, we have removed language that suggests that we are demonstrating the mechanisms in this paper. We have modified this sentence to read “elucidates in more detail the potential impacts of such a strategy.”

- My second major concern is still about the Methods. While the authors have provided much more information regarding the policies evaluated in the Tables and SI that have been added, I’m concerned that the method is not reproducible given the insufficient details provided. For a scientific paper to be published as an article in a high-profile journal like Nature Comms, the method needs to be reproducible. The major part that is still not clear to me is the Athena component. I now understand it much better (see below comment), but it appears to be a decision-making process, rather than a model or tool, to determine which non-Federal actors’ commitments are included or are considered overlapping with state efforts. However, insufficient details are given that would allow for reproducibility of their analysis.

Thank you for the comment. Our responses are described in more detail below; the summary is:

1. To avoid confusion, we have removed the term “Athena” altogether. This was originally an internal term we used to distinguish the process of quantification and aggregation from the mechanical inner workings of GCAM.
2. We now provide the quantifications of the subnational actions at the state-level in our SI. We are confident that given these data and the description of the aggregation process, other researchers can reproduce our results using GCAM or any other integrated assessment model with state level resolution.

References:

Martin, G., & Saikawa, E. (2017). Effectiveness of state climate and energy policies in reducing power-sector CO 2 emissions. Nature Climate Change, 7(12), 912-919.
Meckling, Jonas, Nina Kelsey, Eric Biber, and John Zysman. "Winning coalitions for climate policy." Science 349, no. 6253 (2015): 1170-1171.

Thank you, we have added these references.

Other responses to comments written in green below.

Conceptual comments: The title of the piece is “diversified climate action” but the focus on the paper seems to be primarily on U.S. state- level climate action. Much of the introduction discusses a “groundswell” and “bottom-up” participation of “diversified” climate actors, but from what I can tell in reading both the main article and the SI is that the focus is just on U.S. state-level action and re-enactment of federal climate policy. It is unclear at all if the authors included business commitments and

how they determined whether there were 'binding mechanisms,' legal or otherwise that would produce the same level of stringency for the various scenarios. The authors should be straightforward and be clear about what this article covers - state-level energy policies and ditch the framing in the introduction about diversification of actors, coalitions, etc. because commitments from these actors do not seem to be evaluated in the models or scenarios.

Thank you for this helpful comment. We absolutely agree that we want to be clear about how we are supporting the thesis with the analysis we have undertaken, so we used these comments as a guide to do that. First, this comment underscores that it is important for us to clarify in the text the extent of our analysis. While it is true that our analysis does include an assessment of state-level energy policies, it does go much deeper than this, and this is one reason we feel that it is a helpful and new contribution to the literature. In fact, we have assessed a large number of diverse city, state, and business policies, covering not just energy generation but also transportation, buildings, the land sector, methane, HFCs, and others. A significant data and analytical effort went into assessing how these diverse and overlapping policies would impact overall U.S. emissions. Accordingly, in the text, we have sought to clarify in all instances what we are doing with respect to our analysis.

As one example, we have modified our abstract to provide a clear statement of what work we did: "We aggregated U.S. state, city, and business commitments within a global integrated assessment model (GCAM), to assess how a comprehensive national climate strategy can be built from expanded actions at subnational levels." We also added the following statement at the end of our introduction: "In this paper, we describe how a significant subset of those actors in the United States—including states and cities, and some businesses—are implementing policies that significantly reduce economy-wide emissions, and how expanded action from those actors can provide a higher platform for potential additional Federal action." In doing this we believe there is one contribution of our analysis that we should clarify for the reviewer, and one area that we agree is a legitimate limitation of our analysis.

The first area that we believe was a matter of misunderstanding is that, while we assess all of these diverse actions from cities and states, our methodology does have us aggregate these actions to the state level for purposes of undertaking an economy-wide assessment of emissions impacts.

In other words, **we run a 50-state model in which each state's emissions are driven by a combination of both state-wide, and city policies.** As such, while our analysis ultimately is based on a 50-state assessment, the drivers of action are carefully assessed. **This is the function of the "Athena" model that we describe in the paper and the SI—and which we have undertaken significant revisions in the text to clarify. Athena is the structure within which we carefully eliminate the potential double counting that can arise from having, for example, a city policy that might be partially or fully subsumed by a similar state policy.** In light of the reviewer's comments, we have clarified the language about our methodology, both in the main text and in the methodology section and SI.

In another way, the reviewer is absolutely correct. That is, we do not seek to incorporate actions of entities that are not states, cities, or businesses. For example, the "We Are Still In" coalition includes a number of specific categories of groups, such as tribal groups and universities, that have relatively few policy levers and a relatively small potential impact on emissions. We do not attempt to quantify the impacts of those types of actors. There are two reasons for this. First, there is not usually sufficient baseline and other data for us to make a confident assessment. Second, it is also our view that compared to the city and state policies, the absolute impact of such activities would be small.

Business action is one category that could potentially impact the results, although likely in a smaller way, and we can only partially cover that area. Unfortunately, most business reporting is done on a company-wide rather than a plant-by-plant basis, so the inability to pinpoint geographical location of reductions precludes a confident assessment of many business actions. **However we do evaluate a limited number of actions, which are listed in the summary table in the Methods and elaborated in more detail in the SI. These include 24 utility decarbonization goals, voluntary corporate actions through EPA Gas STAR, and voluntary HFC action through GreenChill. We evaluate these as so-called *pledged actions* and therefore do not include them in our Current Measures scenario. We do include these actions in the more ambitious scenarios, although they are frequently overridden by other, more ambitious assumptions.**

We thus view our assessment as a lower bound on the impact of subnational actions. At the same time we think that given other uncertainties, our numbers are robust. Because we are capturing what we

believe are the most significant actions, we believe that the analysis does effectively illuminate the potential impacts of subnational action on the U.S. national emissions trajectory.

Response:

- *In the text, the clarifications between NSA participation and those actors that are quantified now make this distinction clear.*

Thank you.

- *The authors should certainly include a discussion of the analysis's limitations, particularly as they result from omitting certain policies/actions, or actors, as described above.*

Thank you. We have added a paragraph on limitations of the analysis in the discussion, that includes a reference to this issue. There are also some comments about limitations now added throughout the text.

- *By 'most significant actions,' do the authors mean in terms of emissions impacts, or potential to contribute to national policy?*

We meant emissions impacts, as this is more in line with our analytical contribution. We've clarified it in the text. For example, "with a potentially significant influence on the overall U.S. emissions trajectory."

In this context, Fig. 1 is completely misleading as I don't see any evidence in the SI that tribes, healthcare organizations, faith-based organizations' commitments are actually analyzed.

Thank you for the comment. In this introductory position in the paper, we are focusing primarily on simply conveying the idea—which we think may not be obvious to people not deeply engaged in this field—that there is a broad diversity of action in the U.S. To this end, we believe that the map helps to illustrate a broader concept, which is the breadth of action across the United States, and thus we have retained this figure to help support the motivation for better understanding the diversity of this action. It was placed in the first section that highlights the overall trend of sub-national action and was meant to give the reader a sense of scale. At the same time, we fully agree with the reviewer that we don't want to mislead readers into thinking we analyzed every action by these coalitions. The dots on the figure do represent all actors, and as the reviewer points out correctly, some of these we did not explicitly model. As such we have clarified the purpose of the figure in the manuscript and figure caption. We have added the following clarifications:

Despite the scale of this groundswell of diverse new climate commitments, the central question remains how such coalitions, acting outside national government authorities, can impact overall national emissions trajectories—and, by extension, how broader action across countries can affect the global emissions trajectory. **The research presented here does this by estimating the impact of a subset of the diverse actors shown in Figure 1: the U.S. states and 285 largest cities who have the most impact on overall U.S. emissions.**

And in the caption to Figure 1:

a) States, cities, businesses, universities, faith-based and other organizations now number greater than 3800. **This study assessed the most significant subset of these actors, focusing on states and the 285 largest cities.**

In evaluating the 50 states and 285 largest cities, we found 168 cities with policies that we were able to include. We describe this process in the text, the Methods section, and in the Supplementary Information. To be fully clear, we have also created a new figure, which is Figure 2 in the Supplementary Information, to show the geographic locations of the cities included.

Response:

- *These edits make the selection process much clearer. Am I right in assuming the 50 states and 285 cities were assessed and (I assume) incorporated into the enhanced non-federal scenario, and the 34 states and 168 cities had quantifiable policies were included as part of the current measures scenario? This feels a bit nit-picky, but this could be one thing that's further clarified in the SI in the first section.*

Good to hear that the revisions were helpful. You are correct about both of those. We have clarified in the SI.

There's already a rich literature on state-level climate/energy policies that the authors fail to engage: Rabe, B. G. (2008). States on steroids: the intergovernmental odyssey of American climate policy. *Review of Policy Research*, 25(2), 105-128. Peterson, T. D., & Rose, A. Z. (2006). Reducing conflicts between climate policy and energy policy in the US: The important role of the states. *Energy Policy*, 34(5), 619-631. Goulder, L. H., & Stavins, R. N. (2011). Challenges from state-federal interactions in US climate change policy. *American Economic Review*, 101(3), 253-57.

Thank you for these suggestions. We agree that the literature on state and non-state climate action goes back to important earlier works, and we absolutely want to converse with that earlier literature. In our previous draft, we did refer back briefly to some of this literature; for example, the Rabe 2008 paper was already cited and discussed in that version. Nevertheless, the reviewer's point is well taken. Accordingly, we have expanded the section referring to this literature more to more fully reflect how those ideas relate to our own work. As part of this, we have also integrated discussion of the additional two papers suggested by the reviewer.

We have also enhanced our discussion of the unique character of the particular circumstances in the United States today, with increased international engagement on climate change, severely diminished federal leadership, and an unprecedented level of explicit, bottom-up action on climate in the United States. This circumstance brings into sharper focus and raises the relevance more than ever before of

6 the role that bottom-up actors might play in climate mitigation. It is this emerging circumstance that provides the motivation for this paper.

Response:

- *The context of the circumstances around US climate action is a helpful addition to the framing of the paper, although see my comment above about the paper's overall framing.*

Good to hear that it was helpful.

The authors primarily frame U.S.-state level action as a net-positive, which I think is largely correct. But this is an assumption that merits some further discussion. For instance, the US has also seen some push-back against climate action at the state level (e.g., Ohio's HB 6 law, which subsidizes coal and guts clean energy programs) -- what is the risk of other forms of this kind of push-back based on shifting political leadership? Could state-level action also be weaponized against climate action, if, e.g., there is a 2020 administration committed to ambitious climate action?

This is an incisive point that raises some important political questions, and one that we agree with fully. We have added some text to reflect this issue in the paper, specifically:

It is important to note that while the aggregate effects of sub-national action may serve to build a higher overall level of action in the United States, individual sub-national actions are not always consistent with long-term climate goals or clean energy transitions; they may also lead to conflicts and economic inefficiencies. States have sometimes also opposed federal climate policies through court actions.

Response:

- *These additions in the most recent draft address this comment.*

Great.

Methodological comments:

3

The authors seem to contradict themselves in the section 'Assessing the current and potential impacts of diversified climate action.' Again, see my comment about the conceptual framing above - the authors should just be straightforward and clear - they are using a state-level policy model to assess national-level scenarios. They seem to dance around some of the challenges of fully incorporating 'bottom-up' action in this section in way that suggests that their model does something (i.e., assess "stakeholder

engagement, data collection, aggregation, and modeling” p. 5. Lines 21-23) that other IAMs and GCMs are unable to do, but I don’t see any evidence of that.

7

Thank you for the comment. We addressed the bigger picture framing question above, as well as how our approach does provide a deeper assessment of bottom up action than previous work. There is a second dimension to this comment here that we can respond to here. Our model is really in three parts, and there is a fourth element that the reviewer mentions here. First, we do have a data collection component of the work, which is both challenging but is an essential part of the contribution of this paper. We obtained data on subnational goals, policies, baselines, and targets from a number of sources, including CDP, Under2MOU, C2ES, ICLEI, carbonn, ACEEE, Second Nature, BEA, U.S. EPA, U.S. DOE (including the SLED tool), and others. Data sources are cited in our sector-level descriptions in the Supporting Information. **Second, we then place these inputs in a context of a broader state, national, and global baseline, and then carefully aggregate the impacts on that baseline up to the state level. That is the function of our Athena tool, which is calibrated to an overall GCAM baseline.** Third, we then take these **state level inputs (i.e. the impacts of subnational policies at the state level)** and run them in GCAM to account for economy-wide interactions across the national and global climate, energy, and land systems. **That three part process is novel for IAMs.**

The other dimension mentioned by the reviewer is stakeholder engagement. This did not impact the modeling strategy above, but instead was used to establish our scenarios. Rather than just make assumptions for subnational action, our process engaged with actors to see what they found to be the most plausible policies to expand at the subnational level. We therefore think that adds an important dimension for the integrated roadmap strategy that is exemplified by this research. However, we do not have space to really describe this process in detail for this paper, so we just mention it briefly. It might be valuable to do as a separate white paper or publication.

In response to these helpful comments, we have added additional material on our overall modeling process and the Athena model in particular, as follows:

P.4 - Clarification of methods To address these challenges, we developed a new, comprehensive strategy to assess the scope and **emissions impact of non-federal climate actions in the United States. We do this by integrating data on baselines and sector-specific policies and commitments into an economy-wide estimate of the U.S. emissions trajectory to 2030. The overall process involved steps of data collection, baseline development, aggregation, and modeling, with parallel stakeholder engagement to help develop the policy platform for scenarios (Described further in the Methodology section and in the SI).** The resulting approach combines detailed accounting of bottom-up actions with an integrated assessment model to understand their overall emission impacts.

Response:

- *In the SI, Tables 3 and 4 are helpful in understanding exactly how and where state, city, and companies are factored into different scenarios.*

Great, thank you.

- *The reframing of the overview in the SI is also helpful in clarifying how the analysis process occurred - particularly the definition of Athena as “a new set of bottom-up models (collectively called Athena) to aggregate non-federal climate policies into state or regional activity data for each sector.” It might be helpful to apply that same framing - of Athena as a set of models rather than a single tool in the main body of the text - e.g., on page 6 and at the top of page 16. See below other comments on Athena.*

This is a helpful suggestion for our treatment of Athena in the text and we have implemented it. We have removed reference to Athena and have specified it more clearly in the text. We respond more fully below to your overall comments.

There is mention of an “Athena” tool but it is not at all described in very much detail in the SI (mentioned briefly in a few paragraphs on p. 2), but it’s not clear what “effects of different actions” (p5., line 29) are included. Why not provide a table in the SI with these details to make it 100% transparent and clear what is actually being evaluated? I found this entire paragraph on page 5 about Athena and it’s integration into GCAM extremely vague and unclear.

Thank you for the suggestion. We have added additional material in the SI to clarify what Athena is and how it works, as well as how it interacts with GCAM. In addition, we agree that the goals and purpose of Athena needed to be made more explicit in the main text. Accordingly, we added the following on p. 5: **In assessing actions that impact emissions with overlapping or nested geographical coverage—for example, a city policy on renewables within a state that has a renewables target— there is a need to eliminate double-counting and otherwise account for potential interactions in the areas of overlap. At the same time, it is also important to ensure that baselines and input data across jurisdictions are consistent to ensure that the resulting broader projection is robust. To address these issues, we developed an aggregation methodology (which we call Athena) that draws from a well-established set of inputs from GCAM and then aggregates the effects of diverse climate policies while explicitly considering interactions and avoids double counting within each sector.** Here is some specific language that points to what I mean:

Response:

● *This is a helpful clarification, but again, it would be helpful to ensure Athena is described consistently across the text. It doesn't appear that Athena is a model or tool as it was initially described (or at least how I was picturing it, similar to an IAM or scenario tool). The authors' clarification is helpful, especially because the Athena methodology is not published elsewhere (as is GCAM on the GitHub link provided) nor is it peer-reviewed. From what I can tell from the added description, figure, and table, it appears to be a way of determining which policies to include due to potential overlaps, which then allows for some modification of GCAM baselines. If this is the case - then why present this method as a model or tool or try to frame it as such, and why give it a name? I don't think that giving a methodological process a mythological name like Athena is actually useful - if anything, it obfuscates the purpose of the paper and makes the reader question the validity and credibility of the GCAM scenarios and outputs. The authors should just explicitly say that they instituted a decision-making protocol for aggregation (similar to Kuramochi et al., 2020) and include those decision-points in a diagram (which in my mind should be a flow chart instead of what is shown in Figure 5's oversimplification of what they had written in the previous text "Athena *Delivers appropriate aggregated inputs back to GCAM-AP*").*

These are all helpful points and well-taken. A few responses.

- First, you are right about what Athena does, and we're glad that the earlier revisions were helpful.
- Second, about the name Athena. We agree with your points. None of us has ever been particularly wedded to the name, and in fact it used to be some more awkward acronym like "Aggregation Tracking... and then something spelling H E N A". The reason we used it is that we found in communicating our methods it was simpler to just use a name. However, we agree we don't want to make it seem more magical (or mythical) than it is, which is just a tool to structure a process of aggregation, eliminate double counting and overlaps, and aggregate subnational policy impacts in a way that will mesh with GCAM-USA. It's essentially done via a spreadsheet (although a big one) that helps us ensure we are counting everything correctly. While technically one can still call it a "tool" or a "model", we agree that usually that those terms connote something with more mathematical interactivity under the hood than what we have here. Accordingly, we've removed references to Athena in the main text and have replaced it with your more transparent suggestions above. Here is one example:
 - **Our aggregation process** is first calibrated to the GCAM-AP baselines. **Then, using a structured methodology, we determine what policies are included in our analysis (see SI), what their impact will be, and the degree to which they generate activity that goes beyond what the previous policy architecture would deliver. This allows us to consistently estimate** impacts at the sector level, in terms of activity data appropriate to that sector (for example, TWh of renewable generation from state and municipal policies, number of zero-emission vehicles required to meet state and local procurement, or HFC emissions reductions from state and corporate initiatives)...Using **this interlinked aggregation method with the GCAM-AP model**, we projected emissions impacts under three scenarios.
- Third, about the reproducibility. Reproducing the Athena outputs (or inputs to GCAM) is feasible and not technically complex. We have offered to make our Athena outputs fully available via an excel spreadsheet. These include things like, "how many GW of solar deployment do we expect

in Alabama by 2025, due to additional state and city policies?” Anybody with a spreadsheet and access to publicly available state and city policies can reproduce any of these without resort to a complex model like GCAM. With our documentation of policies we have modeled in SI and the quantification at 50-state level, both the Athena results and our overall results from GCAM are reproducible. We have added some text in the SI to address the question of reproducibility.

- *It seems clearest to me as it's initially described at the start of the SI: as “a new set of bottom-up models (collectively called Athena) to aggregate non-federal climate policies into state or regional activity data for each sector” that's part of the broader 4-step methodology described in the response above.*

Thank you, that's helpful.

“Explicitly considering interactions to avoid double counting” - ?? what do the authors mean by ‘interactions’? It seems they're framing them as negative but they could also be positive (e.g., amplifying, see Chan et al., 2015)?

We address potential issues with double counting in two ways. First, we account for overlap between policies within a sector in Athena (for example, cities within a state that both have a RPS). Second, we model interactions between sectors in GCAM (for example, impacts of electric vehicle policies on electricity demand). We have added further discussion of our process for accounting for overlap in the SI (see *Avoiding Double Counting*). To avoid confusion, we changed “interactions” to “overlap” in the sentence referenced by the reviewer.

We recognize that in practice policies can be overlapping or reinforcing in terms of impact. We discuss how we account for overlap on a sector-by-sector basis in more detail in the *Sector Descriptions* section of the SI. For example, for RPS policies:

We aggregated renewable energy generation resulting from the state, city, and utility actions described above at the state level, partially accounting for overlap across the three different levels of action. We accounted for city policies using a net percentage approach, where additional renewable demand from city goals in percentage terms (beyond the percentage required by state policies) is added to the state total. For example, a city with a 50% goal in a state with a 40% RPS add a net 10% renewable generation to its electricity load. Alternatively, if the city's projected renewable target were less than the state's 40% target, no additional renewable demand would be included. We applied the same net percentage approach when accounting for utility targets.

Response:

- *These changes make the discussion of overlaps and double-counting much clearer. Ideally, the authors could actually include these descriptions and decision-points in a flow-diagram revisioning of Figure 5.*

Thank you for the suggestion and we're glad to know that the revisions were helpful. We have updated Figure 5 and also created a new figure in the SI (Figure 3) that more clearly shows the state aggregation process.

“Delivers appropriate aggregated inputs back to GCAM-AP” - this statement is also too vague and not reproducible.

We agree this statement is vague, and was also unnecessary. We have deleted it.

“Accounting for overlap” - how? What overlaps? Where? Between actors/sectors/emissions?

Here we meant between actors -- for example, a city within a state that both have a renewable portfolio standard. We have clarified this in the text.

Again, ideally the authors could actually include these descriptions and decision-points in a flow-diagram revisioning of Figure 5.

We addressed this question in our response above.

“Athena generates estimates of impacts in terms of activity data appropriate to that sector” - is this done on an actor basis, state-level?

This is done on a sector level. We clarified this.

A diagram that explicitly makes clear the flows between Athena and GCAM would be 100% helpful here. The authors should remove Fig.1 for being misleading and instead actually provide a diagram of the interaction between Athena and GCAM-AP. Also, Athena has not been published or peer-reviewed, from what I could tell from a web of science search, so I question it's rigor or what it's doing, especially because the authors provide scant details on what it includes, how it's doing its "interaction consideration" and "avoiding double counting" and "appropriately aggregating inputs."

Thank you for this suggestion. We have created a new diagram illustrating the modeling process and feedbacks (in the Methods section of the main paper).

Would like to see a clear listing of exactly what policies are included and their estimated emissions reduction impacts. It's unclear from the SI what specific policies are being included and analyzed. In several places, the authors mention "and many other policies," - this is vague and not reproducible.

We have now added a table listing the policies included in each scenario (in the Methods section of the main paper). We describe these policies in more detail in the *Sector Descriptions* section of the SI. Since we model emissions reductions across the U.S., we do not estimate emissions reductions resulting from individual policies. However, we have included a new figure in the main text that breaks down emissions reductions contributions by sector.

Response: The additional tables are extremely helpful.

Great, thank you.

The authors' approach to measuring Tier 1, Tier 2, and Tier 3 states, based on their likelihood of adopting climate action is a potentially very helpful contribution to attempts to predict the impact of climate action in the US. More details on the criteria that informed their classifications (e.g., how exactly, the outlined criteria for placing a state in each tier were defined) could be useful to other analysts focused on non-state climate action.

Thank you for this suggestions, we have added a listing of the criteria in the main text:

In parallel, we generated three groupings of states that reflect the likelihood of accelerated action, based on attributes such as membership in climate organizations, vocal leadership in support of climate action, ambitious emissions reduction targets or standards, and on-the-books climate policies. In our scenario, Tier 1 states realize levels of ambition roughly equivalent to today's leading states like California, Tier 2 states lag but still move toward the level of ambition of Tier 1, and Tier 3 states which do not implement major changes (SI).

Response: This description is helpful; a rubric outlining some of the thresholds and sources for evaluating these criteria would be nice/useful (perhaps something that could be produced as part of future outputs?), but I'm not sure it's necessary.

That's good to have the feedback. We would like to continue building out this methodology in the future and this can be something we focus on for future publications.

The baseline remains unclear - the potential reductions of the current measures, enhanced non-federal measures and comprehensive measures are compared to the NDC target. At times in the SI it seems as if the current measures scenario is used as a kind of baseline, but the implementation of existing subnational commitments is far from certain, and, I would argue, worth understanding on its own terms and for its own contributions from a BAU pathway. The assumptions underlying the trajectory of national emissions in this scenario could be made clearer and more detailed. Page 7 reads: "Figure 2 shows the results of each of the three scenarios. Current Measures would deliver reductions of 25% below 2005 levels by 2030. In contrast, absent any climate policy action, we estimate that absent policies, economic growth would lead to a roughly 4% increase in national emissions in the United States." More details about this baseline scenario (does it project current emissions trends forward based on economic growth? Does it assume current federal policies continue to be implemented?) - in both the text and in Figure 2 would help the reader better understand the potential impact of bottom-up climate action.

We have added a new section to the SI to describe our reference scenario in further detail:

Initial data from GCAM-AP was generally interpreted in Athena as a no-policy, reference scenario in which subnational policies, and some key federal policies, are not represented. Thus, the full impact of policies was applied to the baseline projections without needing to address overlap. Exceptions to this

assumption and cases where any subnational policies were already embedded in the baseline are discussed in the sector descriptions in Section 5.

The GCAM-AP reference case scenario does include certain federal-level policies that have significant impacts within the sectors modeled. These include the federal production tax credit (PTC) and investment tax credit (ITC) in the renewable energy sector and federal fuel economy standards in the transportation sector. While sectoral modeling results in Athena typically represent the impact of state, city, or business policies only, final modeling results from GCAM-AP account for the combined impacts of these federal-level policies and the non-federal impacts from Athena. In addition, two types of federal policies not already included in GCAM-AP were explicitly modeled in Athena and aggregated with non-federal actions before being passed back to GCAM-AP as inputs. These were the U.S. EPA Section 608 refrigerant management policy for HFCs and regulations to reduce fugitive emissions in the oil and gas sector.

Response: This is a helpful clarification. One question about the added text: in the sentence “While sectoral modeling results in Athena typically represent the impact of state, city, or business policies only, final modeling results from GCAM-AP account for the combined impacts of these federal-level policies and the non-federal impacts from Athena” does the text “these federal-level policies” refer to the “federal production tax credit (PTC) and investment tax credit (ITC)”?

Yes, the federal level policies indeed include PTC and ITC.

How do their assessments compare with other assessments of U.S. “enhanced ambition” or re-engagement of the federal government? Would like to see in the discussion some engagement with other analyses and literature (i.e., the Rhodium Group, Kuramochi et al., 2017 and Kuramochi et al., 2019). No references are made to these other pieces in either the main text or the sensitivity analysis that also analyze future potential emissions reductions from U.S. subnational/non-state actors, which are very similar in their analyses.

There have been a few other assessments of the impact of subnational action on emissions both at the global level and, in a couple of cases, in the United States specifically. We discussed the global perspectives in our framing sections, but the reviewer makes a good point about two analyses that look specifically at U.S. emissions under current measures. Kuramochi and others are co-authors on the two studies that we noted in the first draft, and we have now added explicit reference paper suggested by the reviewer as well as one new one. Substantively, this group is strong and we agree they have done good work looking at global impacts in particular. Because their focus is global, their assessment of the U.S. is necessarily less detailed. Their numbers are roughly in line with ours, but their aggregate approach does not allow for the explicit linking of policy roadmaps that our more detailed analysis allows. In this sense, it remains our view that the methodological approach we present here is an important advancement. Also, our are broadly consistent with Rhodium Group’s study from 2017, although their lack of published methodologies makes a detailed comparison of comparability between their results on ours somewhat challenging. We have added a reference to this in our paper, because we agree with the reviewer that some comment is helpful here. We have added the following to the text:

There is currently no published estimate directly comparable to our higher-ambition *Enhanced Non-Federal* and *Comprehensive* scenarios. On the other hand, the estimates for the impact of existing policies, including state and city actions (i.e., the *Current Measures* scenario), are broadly consistent with coarser estimates that have been made for the potential of U.S. emissions reductions. Kuramochi et al., for example, assessed the impact of current climate policies on 25 key emitters, including the United States. Their global assessment was necessarily less detailed in its roadmap for how climate action could be implemented in the U.S. from sub-national action, and the best analogue to our assessment is to our *Current Measures* projection. Their range of outcomes in 2030 shows a trajectory for the U.S. to achieve roughly a 15-26% reduction, consistent at the upper end with our central estimate of 25%. Similarly, in 2017, the Rhodium Group used publicly available versions of GCAM and NEMS to forecast near-term U.S. emissions, and they projected a 15-18% reduction in 2025 and a 14-19% reduction in 2030, relative to 2005. Our *Current Measures* results are slightly higher than these, which we attribute to the fact that those estimates do not fully include the sub-national actions that our estimate includes, and they also pre-date much of the recent wave of bottom-up climate commitments described earlier

Response:

- The added text addresses the original comment.

Great

Other notes:

Small typo in "Paris Agreement" (Page 4, line 16)

Thank you for catching this. This is now fixed.

On page 7: "In contrast, absent any climate policy action, we estimate that economic growth would lead to a roughly 4% increase in national emissions in the United States." Am I right in assuming this means absent any additional climate policy action?

Yes, indeed this is from a counterfactual scenario in which no additional climate policy action occurs. However, we ended up removing this sentence altogether in the revision. We didn't find that predicting the impact of economic growth in this particular environment was particularly useful for communication purposes. We plan on taking up on this particular issue in our upcoming publications.

Reviewer #2 (Remarks to the Author):

Overall: We appreciate the contributions from Reviewer 2 to support our earlier revision. There are no new comments to address.

This paper discusses a trending issue to provide the possible mitigation sub-national actors might deliver. The modeling based on GCAM USA and Athena provides reliable quantification of how these sub-national actors can contribute to the climate actions.

Thank you for the comment and reflection on the relevance and reliability of our analysis.

However, the detail has not been clarified enough for the reader to understand how many firms or cities have been involved in this study, what kind of action has the best effect, etc.

We have provided more detail about this now in the paper, and have enhanced our descriptions of this wherever possible.

The authors raised two interesting questions in the beginning, on 'What bottom-up actor groups could deliver? And how these groups could support greater national actions?' I think more results could be included to discuss these two questions and your findings.

Thank you for this comment. We have added some additional results into the main text, including a new figure, to provide additional context for and support of our argument.

My detailed comments are listed below

Page 4 line 9: How do you eliminating overlap for the population and GDP? The number seems very impressive that 65% of the US population has been engaged in climate actions, but I'm not very certain if you can detect if one person been engaged in several sub-national coalitions? (I've also roughly go through the report you cited, perhaps I have missed it, but didn't see how they eliminate overlap)

Here we are calculating the percent of the U.S. population whose subnational governments have declared support for the Paris Agreement. We avoid double counting by not counting the population of any cities within a state that has also declared support. We have added further clarification in the text.

Page 5 line 29: I would suggest you include more detail on the Athena model if this is one of your innovations. The underlying theoretical mechanism needs to be explained here. And I also recommend you to include 'Figure 2 ATHENA Modeling Flow' from your technical appendix of 'Fulfilling America's Pledge' in the supplementary materials.

We have added a paragraph in the main text describing Athena, and have improved and included the figure suggested (Figure 2 Athena modeling flow). It is now in the Methodology section of the main text.

Page 6: The name of the scenario could be better unified long the text (Bottom-up/All-in vs Enhanced non- federal/Comprehensive).

Thank you, we have updated all of our scenario names to be consistent.

Page 6 Scenarios: It would be nice to clarify the scenario difference more, either in the explanation of scenario setting or in the supplementary material.

We have added substantial elaboration on the scenarios in the main text Methods and in the SI. We describe the conceptual differences in the main text and in bullets in the Methods:

- *Current Measures projects where the U.S. is headed given current policies at national, state, and local levels (Table 1).*
- *Enhanced Non-Federal examines outcomes under a broad expansion of cutting-edge climate policies on the part of states, cities, and businesses (Table 2).*
- *Comprehensive explores a climate strategy integrating aggressive non-federal climate action with renewed federal engagement after 2020 (Table 2).*

Table 1 and 2 provide descriptions of the specific actions included in our analysis. We also included more descriptions of our methods in the new *Scenario Development* section of the SI as well as in the *Scenario Descriptions*.

Page 6 line 18: For enhanced non-federal scenario, I'm a bit confused about why you classify the level of potential state ambition and how it works for Athena.

We group states in different tiers to reflect their willingness to engage in climate action. We identified these states by attributes such as membership in climate organizations, vocal leadership in support of climate action, ambitious emissions reduction targets or standards, and on-the-books climate policies. We added more detail on this in the main text when we describe the *Enhanced Non-Federal* scenario.

Page 7 Figure 2: How do you address the uncertainty for the three scenarios? Why the emission trajectory (Figure 2) have a range of uncertainty but the sectoral results (Figure 3) don't have?

We added further discussion of the treatment of sensitivities in the main text:

To better characterize the range of possible emissions reductions of the scenarios in this study, we examined each scenario under varying assumptions about socioeconomic change, technological change, fossil fuel prices, and the size of the land sink. Based on these sensitivities, reductions in the Comprehensive scenario could be as low as 46% or as high as 52%. While these are only a subset of uncertainties, they provide a range of possible emissions associated with different levels of climate action. (See Methods for further discussion.)

We also add discussion in the Methods:

We also examine economy-wide results across a range of values for important drivers in our analysis. These include economic growth, population growth, oil and gas prices, renewable energy costs, coal power plant retirements, and the magnitude of the land sink. While these represent a subset of all possible uncertainties, they give a partial indication of the range of possible emissions for each scenario in our analysis. See Table 2 in the supporting information for a list of assumptions and sensitivities for the economy-wide analysis.

We further discuss these in the SI. For visual simplicity, we include our sensitivities in our line graph showing emissions reductions but not in the sectoral results.

Page 8 Figure 3: Could be better using transparency to differentiate the historical data and model results for Figure 3 (a). We added clarification in the caption that the grey line indicates historical data.

Page 9 line 30: 'The next year' seems to be quite rough. It would be nice to notice which year it is (as I suppose is this year, 2020, now?).

We have changed the text to "This is a critical period" - given that now the timeline for submitting new NDCs may spill into next year, but no decisions have been made yet, we thought it best not to anchor to a specific year.

Page 9 Discussion: You made a very strong argument on why we need to focus on the sub-national actions and I totally agree these were more important than ever for the time being. Yet I would suggest you include more findings from your model to support this idea, with such a big model like GCAM, it would be such a loss to show only such few results.

Thank you for the comment. The reviewer is correct that GCAM produces a vast quantity of outputs, many of which are interesting and worth discussing. So we are happy to have the invitation to share more results. Mindful that we still have a space constraint, we have added one additional figure and some associated text: "we have also prepared the sectoral policy numbers by state that serves as a building block to the state-level modeling. These files are attached for the review, and will be made available upon request (due to the inclusion of proprietary data)."

This figure shows a sectoral breakdown of where the emissions reductions come from within our model framework. Combined with the deeper dive on the power sector, we think this gives a more complete picture of our results.

REVIEWERS' COMMENTS:

Reviewer #1 (Remarks to the Author):

The authors' revisions to the paper addresses my main comments:

The transparency of the methodology (particularly around the processes formerly described under the "ATHENA" heading, and the ways in which city and company actions are factored into the modeling)

The framing of the article's main focus (on estimating the mitigation impact of specific bottom-up contributions, and whether they could be used to achieve deep emissions reductions) governance processes to scale up responses to climate change.

I applaud the authors for addressing my comments in detail and for reframing the paper - I really think it is much clearer and compelling.

I do have one additional comment, however, that I think would need to be addressed before the paper can be published, which is in the Discussion section (p.11 lines 23:34). This paragraph is pretty speculative and overly optimistic of states' abilities (and motivations) to "generate [sic] more diversified policies and commitments [that] stimulate broader engagement for decisions that are more directly tied to constituents." While I don't disagree with this sentiment, the analysis presented in the paper doesn't provide any evidence of these trickle-down effects on constituents (this goes back to my original comment about the lack of demonstrating mechanisms/causal chains). In fact, I think there's evidence of the contrary, where diverse coalitions (i.e., lobby groups, utilities) have acted counter to ambitious climate policies despite having state-level policies like RPS and net-metering in place (Leah Stokes' book "Short Circuiting Policy" provides four examples where this happens - Texas, Ohio, Kansas and Arizona). I know these states are likely in the Tier 3, but she provides evidence of where states had the right clean energy/climate policies in place, but they failed to generate broad local support or stickiness because of interest groups and lobbies that worked to gradually reduce the scale of these policies. This article, although its main goal is to show the potential for scaling up state policies' impact on climate mitigation, also needs to speak to this realism in the discussion section and as a limitation (which right now seems to focus on the data).

I also greatly appreciate the authors' offer to open up the ATHENA spreadsheet/tool - I really think it will be a huge asset to both the policy and research communities to be able to adapt.

A few minor points are listed below for the authors' consideration:

- I might suggest considering the term "non-state" rather than "subnational" to replace "diversified," since there's a bit more precedent around using non-state to describe both government and company actions - but it is clear from the text what kinds of actors (cities, companies, and states) are being evaluated, and otherwise these changes in the title and framing make the focus of the paper much clearer.
- In the SI section 2 on scenario development, suggest noting which data sources were examined to determine company commitments.
- In Table 1, for vehicle emissions standards, it would be helpful to note how many or which states "adopt California's clean cars compromise ensuring incremental vehicle improvement through 2025."
- In Table 1, small typo in the Regulation of Fugitive Oil and Gas Operations" row - "for new or new"

Response to Referees NCOMMS-20-08589

“Fusing subnational with national climate action is central to rapid decarbonization: The case of the United States,” Hultman et al.

July 27, 2020

Updated Aug 12, 2020

Reviewer #1 (Remarks to the Author):

The authors’ revisions to the paper addresses my main comments:

The transparency of the methodology (particularly around the processes formerly described under the “ATHENA” heading, and the ways in which city and company actions are factored into the modeling) The framing of the article’s main focus (on estimating the mitigation impact of specific bottom-up contributions, and whether they could be used to achieve deep emissions reductions) governance processes to scale up responses to climate change.

I applaud the authors for addressing my comments in detail and for reframing the paper - I really think it is much clearer and compelling.

Thank you for your helpful comments.

I do have one additional comment, however, that I think would need to be addressed before the paper can be published, which is in the Discussion section (p.11 lines 23:34). This paragraph is pretty speculative and overly optimistic of states’ abilities (and motivations) to “generate [sic] more diversified policies and commitments [that] stimulate broader engagement for decisions that are more directly tied to constituents.” While I don’t disagree with this sentiment, the analysis presented in the paper doesn’t provide any evidence of these trickle-down effects on constituents (this goes back to my original comment about the lack of demonstrating mechanisms/causal chains). In fact, I think there’s evidence of the contrary, where diverse coalitions (i.e., lobby groups, utilities) have acted counter to ambitious climate policies despite having state-level policies like RPS and net-metering in place (Leah Stokes’ book “Short Circuiting Policy” provides four examples where this happens - Texas, Ohio, Kansas and Arizona). I know these states are likely in the Tier 3, but she provides evidence of where states had the right clean energy/climate policies in place, but they failed to generate broad local support or stickiness because of interest groups and lobbies that worked to gradually reduce the scale of these policies. This article, although its main goal is to show the potential for scaling up state policies’ impact on climate mitigation, also needs to speak to this realism in the discussion section and as a limitation (which right now seems to focus on the data).

Thank you for that comment. We have rewritten that paragraph to first highlight the genuine opportunity for non-state action and then note that the mechanisms for achieving it are complex, subject to political blocking (referencing Stokes) and require further study. We think this eliminates what could have been interpreted as speculative language.

I also greatly appreciate the authors’ offer to open up the ATHENA spreadsheet/tool - I really think it will be a huge asset to both the policy and research communities to be able to adapt.

A few minor points are listed below for the authors’ consideration:

- I might suggest considering the term "non-state" rather than "subnational" to replace "diversified," since there's a bit more precedent around using non-state to describe both government and company actions - but it is clear from the text what kinds of actors (cities, companies, and states) are being evaluated, and otherwise these changes in the title and framing make the focus of the paper much clearer.

Thank you. Yes, our primary hesitation in using "non-state" had been the implication that it was not governments—and some of our key policies are from states and cities-- but it's a helpful point that the general usage is less precise. Accordingly, we have added and substituted "non-state" in a few places to help minimize distraction from the central argument. For a few places where "subnational" is still clearer or more appropriate, we have left it in. (For example, it's confusing to say "state and other non-state actions" instead of "state and other subnational actions").

- In the SI section 2 on scenario development, suggest noting which data sources were examined to determine company commitments.

We have noted in that section that we use utility-level commitments sourced from the Smart Electric Power Alliance (SEPA) tracker, EPA Natural Gas STAR, and EPA GreenChill as data sources.

- In Table 1, for vehicle emissions standards, it would be helpful to note how many or which states "adopt California's clean cars compromise ensuring incremental vehicle improvement through 2025."

The bottom-up scenario assumes "all states adhere to the California compromise through 2026." (Appendix A, Table 11). California and Tier 1 and 2 states move forward with ambitious standards from 2026-2030.

- In Table 1, small typo in the Regulation of Fugitive Oil and Gas Operations" row - "for new or new"

Thank you, we have fixed this.